# Diffusion markers of dendritic density and arborization in gray matter predict differences in intelligence

Erhan Genç[1], Christoph Fraenz[1], Caroline Schlüter[1], Patrick Friedrich[1], Rüdiger Hossiep[2], Manuel C. Voelkle[3], Josef M. Ling[4], Onur Güntürkün[1,5] & Rex E. Jung[6]

Previous research has demonstrated that individuals with higher intelligence are more likely to have larger gray matter volume in brain areas predominantly located in parieto-frontal regions. These findings were usually interpreted to mean that individuals with more cortical brain volume possess more neurons and thus exhibit more computational capacity during reasoning. In addition, neuroimaging studies have shown that intelligent individuals, despite their larger brains, tend to exhibit lower rates of brain activity during reasoning. However, the microstructural architecture underlying both observations remains unclear. By combining advanced multi-shell diffusion tensor imaging with a culture-fair matrix-reasoning test, we found that higher intelligence in healthy individuals is related to lower values of dendritic density and arborization. These results suggest that the neuronal circuitry associated with higher intelligence is organized in a sparse and efficient manner, fostering more directed information processing and less cortical activity during reasoning.

[1] Institute of Cognitive Neuroscience, Biopsychology, Department of Psychology, Ruhr University Bochum, 44801 Bochum, Germany. [2] Team Test Development, Department of Psychology, Ruhr University Bochum, 44801 Bochum, Germany. [3] Psychological Research Methods, Department of Psychology, Humboldt University Berlin, 10099 Berlin, Germany. [4] The Mind Research Network/Lovelace Biomedical and Environmental Research Institute, Albuquerque, NM 87106, USA. [5] Stellenbosch Institute for Advanced Study (STIAS), Wallenberg Research Centre at Stellenbosch University, Stellenbosch 7600, South Africa. [6] Department of Neurosurgery, University of New Mexico, Albuquerque, NM 87131, USA. These authors contributed equally: Erhan Genç, Christoph Fraenz. Correspondence and requests for materials should be addressed to E.G. (email: erhan.genc@rub.de)

Individuals differ with regard to their intellectual abilities in a manner consistent with a normal distribution. The measure most commonly used to quantify broad mental capabilities of an individual is that of intelligence, often termed the intelligence quotient (IQ). Over the last century, researchers have constructed a large number of psychometric test batteries targeting various realms of intelligence. This line of research has created a foundation for quantifying interindividual differences in intelligence with both high reliability and validity[1]. It has also revealed the importance of intelligence for predicting various aspects of everyday life, including academic success, professional advancement, social mobility, physical well-being, and even life expectancy[2].

From the very beginning of intelligence research, there has been a profound interest in linking interindividual differences measured by psychometric test instruments to differences possessing a neurobiological substrate. Early attempts relating brain size to intelligence can be traced to the nineteenth century, with scientists including Pierre Paul Broca and Francis Galton demonstrating a positive relationship between coarse measures of head size and intellectual ability[1,3]. Contemporary differential psychologists have fully embraced the possibilities of neuroscientific methods, especially magnetic resonance imaging (MRI) techniques. Over several decades, a large body of evidence has consistently shown that bigger brains tend to perform better at tasks related to intelligence. Meta-analyses have reported correlation coefficients in the range of 0.24–0.33 for the association between overall brain volume and intelligence[4,5]. This moderate structure–function relationship can be observed for the whole brain, its lobar volumes, and even within specific brain areas predominantly located in parieto-frontal regions[6]. A common biological explanation for this association is the fact that individuals with more cortical volume are likely to possess more neurons[7,8] and thus more computational power to engage in problem solving and logical reasoning.

In the late 1980s, researchers made an important contribution with regard to the biological bases of intelligence, namely, the first PET scan conducted while performing the Raven's Advanced Progressive Matrices[9]. They found a negative correlation between Raven scores and absolute regional metabolic rates, suggesting lower energy consumption in individuals with higher Raven scores[10]. This study was the first to hypothesize that intelligence is not a function of how hard the brain works but rather how efficiently it works, an observation known as the neural efficiency hypothesis of intelligence[11,12]. The hypothesis that intelligence is accomplished through efficient rather than excessive information processing by the brain's neuronal circuitry has been supported by several studies using a wide range of neuroscientific methods[12]. Thus, the notion that intelligence is largely determined by brain size has been criticized for being far too simplistic. A more recent working hypothesis endorses the idea that interindividual differences in intelligence are, to a significant extent, manifested in the wiring properties of brain tissue, for example, in circuit complexity or dendritic arborization[13].

Evidence supporting the neural efficiency hypothesis of intelligence mainly comes from studies investigating brain function by the use of PET, fMRI, and EEG methods[12]. Apart from a few post mortem examinations, little is known about the anatomical substrates of neural efficiency[14]. This is due to a lack of practical in vivo methodologies to examine the microstructural correlates of efficient information processing at the level of axons or dendrites. Currently, the most promising technique for the quantification of neurite morphology is a diffusion MRI technique known as neurite orientation dispersion and density imaging (NODDI). This technique is based on a multi-shell high-angular-resolution diffusion imaging protocol and offers a novel way to analyze diffusion-weighted data with regard to tissue microstructure. It features a three-compartment model distinguishing intra-neurite, extra-neurite, and cerebrospinal fluid (CSF) environments. NODDI is based on a diffusion model that was successfully validated by histological examinations utilizing staining methods in gray and white matter of rats and ferrets[15,16]. In addition, Zhang, Schneider[17] have shown that NODDI is also capable of estimating diffusion markers of neurite density and orientation dispersion by in vivo measurements in humans. Direct validation of NODDI has recently been performed in a study investigating neurite dispersion as a potential marker of multiple sclerosis pathology in post-mortem spinal cord specimens[18]. The authors reported that neurite density obtained from NODDI significantly matched neurite density, orientation dispersion, and myelin density obtained from histology. Furthermore, the authors also found that NODDI neurite dispersion matched the histological neurite dispersion. This indicates that NODDI metrics are closely reflecting their histological conditions.

Here we present the first study using NODDI to examine the microstructural fiber architecture of the human brain in order to shed light on possible neuroanatomical correlates affecting intelligence. We demonstrate that NODDI measures of neurite density and arborization show negative relationships to measures of intelligence, implicating neural efficiency, particularly within parieto-frontal brain regions, as suggested by the vast majority of neuroimaging studies of intelligence[6,19,20].

## Results

**Associations on a whole-brain level.** All analyses were performed with data from two independent samples, namely, an experimental sample (S259) and a validation sample (S498). In the experimental sample we included healthy participants ($N = 259$, 138 males) between 18 and 40 years of age (M = 24.31, SD = 4.41). We determined macrostructural and microstructural brain properties and examined their relationship with cognitive measures of intelligence. Intelligence was assessed with a matrix-reasoning test called Bochumer Matrizentest (BOMAT)[21]. The BOMAT test scores ranged from 7 to 27 correctly answered items (M = 15.75, SD = 3.72) with 28 items being administered in total. We examined brain macrostructure via cortical volume ($VOL_{Cortex}$) and overall white matter volume ($VOL_{WM}$) (Fig. 1, right box) by using an automated brain segmentation procedure[22,23] on the participants' high-resolution anatomical scans. Brain microstructure was quantified with NODDI coefficients representing neurite density, neurite orientation dispersion, and isotropic diffusion within the cortex ($INVF_{Cortex}$, $ODI_{Cortex}$, $ISO_{Cortex}$) and white matter ($INVF_{WM}$, $ODI_{WM}$, $ISO_{WM}$)[17,24] (Fig. 1, right box). For the purpose of validating our experimental results, we used data provided by the Human Connectome Project[25]. This sample included 498 participants (202 males) between 22 and 36 years of age (M = 29.16, SD = 3.48). As with sample S259, the intelligence test scores from sample S498 were also obtained with a matrix-reasoning test, in this case the Penn Matrix Analysis Test (PMAT24)[26]. The PMAT24 test scores ranged from 5 to 24 correctly answered items (M = 16.53, SD = 4.74) with 24 items being administered in total. The neuroimaging data from sample S498 were processed identically to sample S259.

In sample S259, significant structure–function associations were observed on a whole-brain level for most of the macrostructural and microstructural brain properties (Fig. 2 and Supplementary Fig. 1). Partial correlations, controlling for age and sex, showed that intelligence was negatively associated with $INVF_{Cortex}$ ($r = -0.13$, $p < 0.05$) and $ODI_{Cortex}$ ($r = -0.21$, $p$

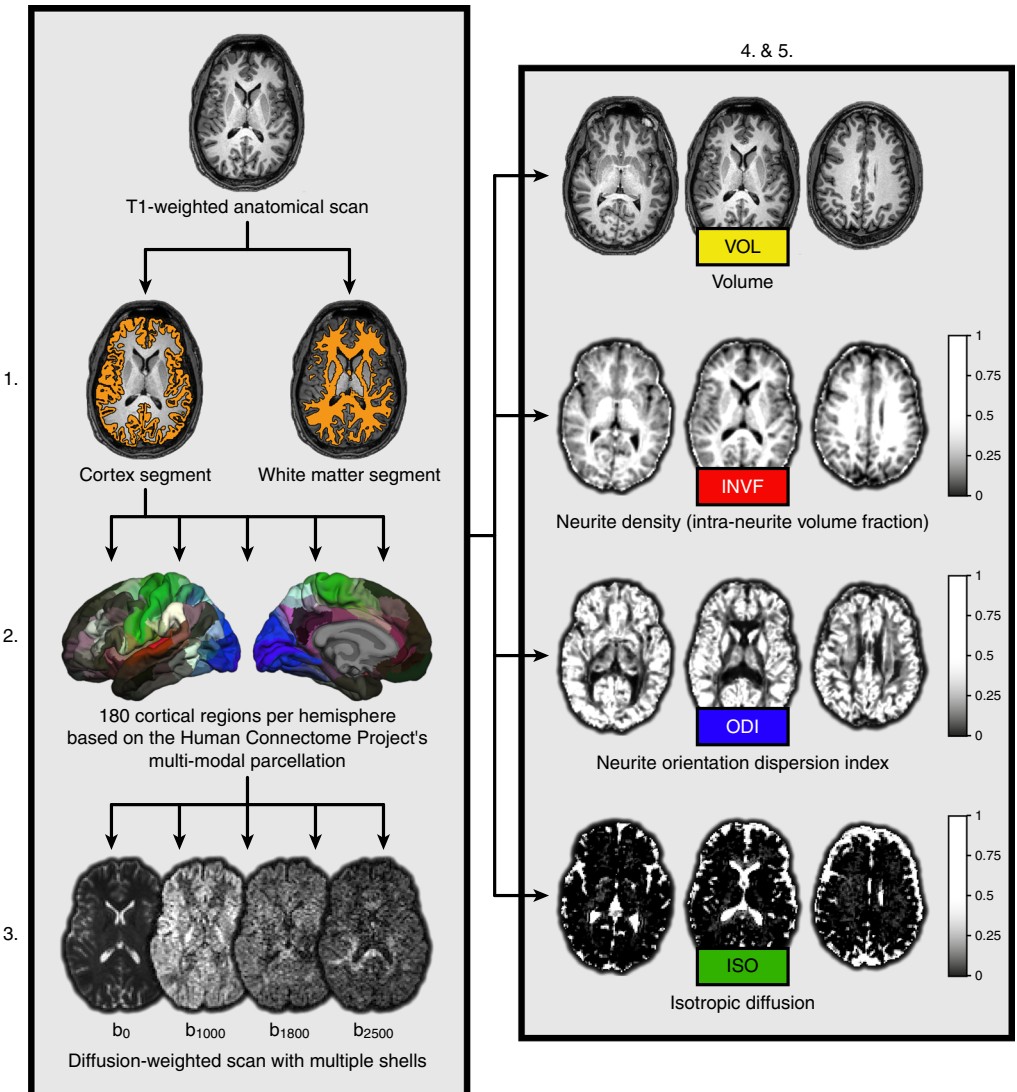

**Fig. 1** Methodological sequence for the estimation of brain properties. First, T1-weighted anatomical images were partitioned into two segments including the overall cortex and white matter of the brain, respectively. Second, the cortical segment was further partitioned into 180 regions per hemisphere based on the multi-modal parcellation scheme provided by the Human Connectome Project. Third, both segments and the cortical brain regions were linearly transformed into the native space of the diffusion-weighted NODDI images. Fourth, mean values of different macrostructural and microstructural measures (volume estimates and NODDI coefficients) were computed for the overall cortex and white matter using the respective segments. Fifth, at the level of single brain regions, volume estimates and NODDI coefficients from homotopic brain regions were averaged across both hemispheres resulting in 180 mean values for each macrostructural and microstructural measure, respectively

$< 0.01$) (Fig. 2), indicating that individuals with less neurite density and less neurite orientation dispersion in the cortex performed better on the intelligence test. Intelligence was not significantly associated with $INVF_{WM}$, $ODI_{WM}$, $ISO_{Cortex}$, and $ISO_{WM}$ (Fig. 2 and Supplementary Fig. 1). Partial correlation analysis showed a significant positive association between intelligence and $VOL_{Cortex}$ ($r = 0.20$, $p < 0.01$) (Supplementary Fig. 1). This result is consistent with previous research linking intelligence with brain size[5,27,28]. However, in contrast to previous research[28], intelligence was not significantly related to $VOL_{WM}$. The results obtained from sample S498 replicated those obtained from sample S259. Partial correlations, controlling for age and sex, revealed that intelligence was negatively associated with $INVF_{Cortex}$ ($r = -0.10$, $p < 0.05$) and $ODI_{Cortex}$ ($r = -0.15$, $p < 0.01$) (Supplementary Fig. 2) and positively associated with $VOL_{Cortex}$ ($r = 0.19$, $p < 0.01$) (Supplementary Fig. 3). Again, intelligence was not significantly associated with $INVF_{WM}$,

$ISO_{Cortex}$, and $ISO_{WM}$, while partial correlation analysis revealed significant negative associations between intelligence and $ODI_{WM}$ ($r = -0.12$, $p < 0.01$) as well as intelligence and $VOL_{WM}$ ($r = 0.10$, $p < 0.05$).

Importantly, the brain properties included in this study are significantly correlated with one another (Supplementary Tables 1 and 2). In sample S259, this is particularly apparent for the association between gray and white matter estimates: $INVF_{Cortex}$ and $INVF_{WM}$ ($r = 0.60$, $p < 0.01$), $ODI_{Cortex}$ and $ODI_{WM}$ ($r = 0.47$, $p < 0.01$), $ISO_{Cortex}$ and $ISO_{WM}$ ($r = 0.71$, $p < 0.01$), as well as $VOL_{Cortex}$ and $VOL_{WM}$ ($r = 0.75$, $p < 0.01$). Therefore, it is reasonable to assume that these brain properties share some of the explained variance when predicting intelligence. Previous research has shown that intelligence and cerebral cortex volume are negatively associated with age[29,30]. This is consistent with sample S259 showing a negative correlation between age and intelligence ($r = -0.17$, $p < 0.01$) as well as age and $VOL_{Cortex}$

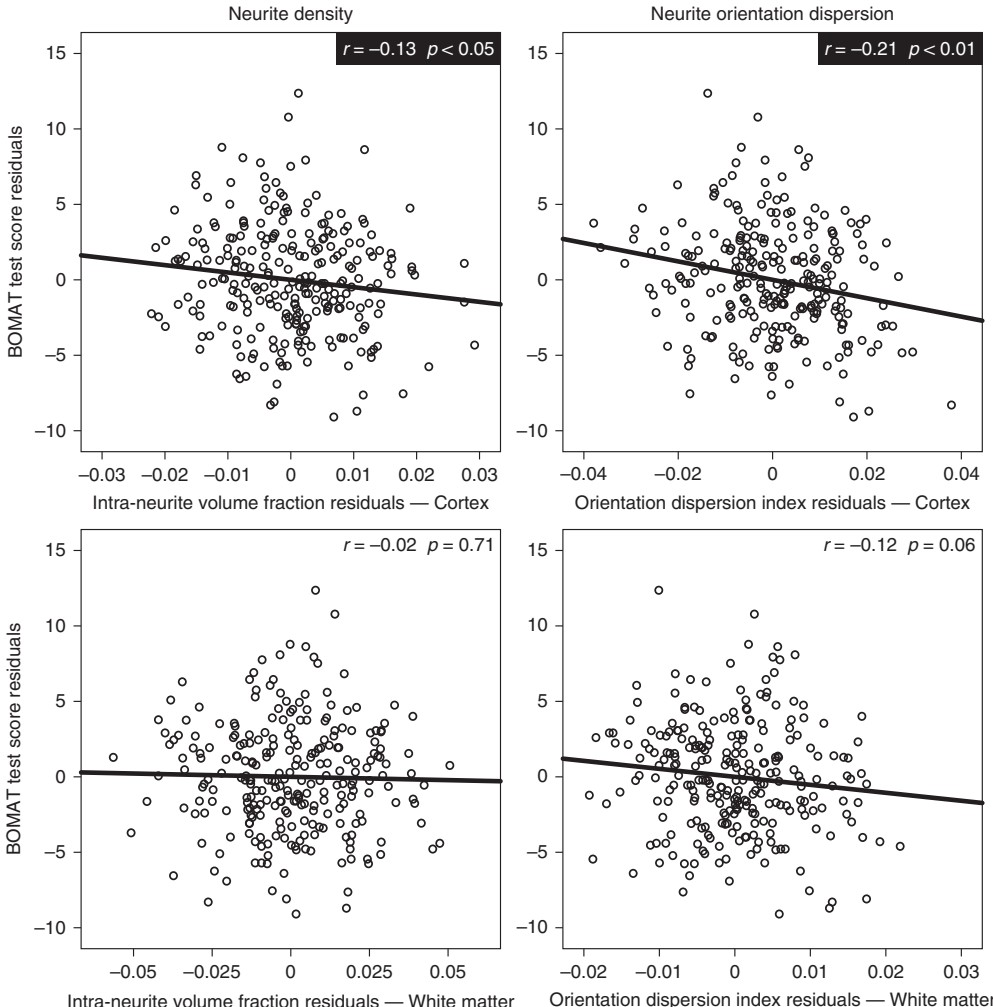

**Fig. 2** Partial correlation analyses with data from sample S259 quantifying structure–function associations at the whole-brain level. Scatter plots illustrating the relationship between neurite density and intelligence are depicted in the left column. Scatter plots illustrating the relationship between neurite orientation dispersion and intelligence are depicted in the right column. In all cases, microstructural measures were computed as mean values derived from the overall cortex (upper row) or white matter (lower row), respectively. Results are based on partial correlation analyses with age and sex being used as controlling variables. Statistically significant partial correlation coefficients ($N = 259$, $p < 0.05$) are highlighted with black boxes

($r = -0.24$, $p < 0.01$). Furthermore, we observed significant sex differences with males having greater $VOL_{Cortex}$ ($t(257) = 10.01$, $p < 0.01$) and $VOL_{WM}$ ($t(257) = 10.63$, $p < 0.01$) as well as higher $INVF_{Cortex}$ ($t(257) = 2.70$, $p < 0.01$) and $INVF_{WM}$ ($t(257) = 2.44$, $p < 0.05$) relative to females. This suggests that the prediction of intelligence by macrostructural and microstructural brain properties might, in part, be confounded by an individual's age and sex or collinearity among the predictors. To address these issues, we employed an approach similar to recent studies investigating the relationship between different brain properties and intelligence[31,32]. We computed a multiple regression analysis that enabled us to extract the unique contribution of each macrostructural and microstructural brain property in predicting intelligence.

In this model, intelligence was regressed on age, sex, and all brain properties included in the partial correlation analysis. The regression model for sample S259 was significant ($R^2 = 0.14$, $F(10, 248) = 3.86$, $p < 0.01$) and yielded significant regression coefficients for $INVF_{Cortex}$ ($\beta = -0.22$, $p < 0.05$) and $ODI_{Cortex}$ ($\beta = -0.19$, $p < 0.05$). The regression coefficient for $VOL_{Cortex}$ was

**Table 1 Summary of multiple regression analysis for variables predicting BOMAT test scores ($N = 259$, $R^2 = 0.14$)**

| Variable | B | SE B | β |
|---|---|---|---|
| $INVF_{Cortex}$ | −70.95 | 31.52 | −0.22* |
| $INVF_{WM}$ | 27.71 | 15.22 | 0.15 |
| $ODI_{Cortex}$ | −55.02 | 22.44 | −0.19* |
| $ODI_{WM}$ | 11.74 | 33.07 | 0.03 |
| $ISO_{Cortex}$ | −8.26 | 13.51 | −0.06 |
| $ISO_{WM}$ | 41.57 | 32.93 | 0.11 |
| $VOL_{Cortex}$ in cm³ | 0.02 | 0.01 | 0.22 |
| $VOL_{WM}$ in cm³ | 0.00 | 0.01 | −0.05 |
| Age in years | −0.02 | 0.06 | −0.03 |
| Sex | 0.06 | 0.57 | 0.01 |

$INVF_{Cortex}$ = intra-neurite volume fraction representing neurite density in the cortex, $INVF_{WM}$ = intra-neurite volume fraction representing neurite density in the white matter, $ODI_{Cortex}$ = orientation dispersion index of neurites in the cortex, $ODI_{WM}$ = orientation dispersion index of neurites in the white matter, $ISO_{Cortex}$ = isotropic diffusion in the cortex, $ISO_{WM}$ = isotropic diffusion in the white matter, $VOL_{Cortex}$ = cortical volume, $VOL_{WM}$ = white matter volume; Sex was represented as a dummy variable with males being labeled 0 and females 1; *$p < 0.05$

of comparable magnitude but failed to reach statistical significance ($\beta = 0.22$, $p = 0.08$) (Table 1 and Supplementary Fig. 4). Nevertheless, these results generally confirmed the pattern revealed by the partial correlation analysis and indicate that the two microstructural brain properties, $INVF_{Cortex}$ and $ODI_{Cortex}$, contribute to the prediction of intelligence independently. Furthermore, we observed no significant associations between intelligence and the remaining predictors $ISO_{Cortex}$, $INVF_{WM}$, $ODI_{WM}$, $ISO_{WM}$, $VOL_{WM}$, age, and sex. It is conceivable that intelligence might be associated with study compliance in such a way that low-IQ individuals show more unwanted head movements during the MRI examination. This in turn might distort the estimated magnitude of certain brain properties and hence affect the outcome of the aforementioned multiple regression analysis. However, in the S259 sample, intelligence was not significantly correlated with head motion measured during the diffusion-weighted scan ($r = -0.03$, $p = 0.69$). Consequentially, adding head motion as a covariate to the multiple regression analysis did not alter the respective results in any substantial way (Supplementary Table 3).

Results of the same regression analysis for sample S498 ($R^2 = 0.08$, $F(10, 487) = 4.27$, $p < 0.01$) were comparable to sample S259 (Supplementary Table 4 and Supplementary Fig. 5). Importantly, we observed significant regression coefficients with a negative sign for $INVF_{Cortex}$ ($\beta = -0.15$, $p < 0.05$) and a positive sign for $VOL_{Cortex}$ ($\beta = 0.27$, $p < 0.01$). $ISO_{Cortex}$, despite not showing a significant correlation with intelligence ($r = 0.02$, $p = 0.62$), had a positive $\beta$ coefficient that reached statistical significance ($\beta = 0.17$, $p < 0.01$). This condition, in which an independent variable shows no correlation with the dependent variable, but makes a significant contribution in the context of a multiple regression analysis with other variables, is called "suppression" in statistics[33–35]. The variable suppresses variance that is not related to the dependent measure in other independent variables and thereby enhances predictive power of the variable set as a whole[36]. Thus, only $INVF_{Cortex}$ and $VOL_{Cortex}$ can be regarded as uniquely contributing to the prediction of intelligence in the S498 sample. None of the remaining regression coefficients reached statistical significance.

**Associations on the level of single brain regions**. Next, we focused our analysis on NODDI coefficients derived from single brain regions in order to draw a more refined picture of the structure–function relationships observed at the whole-brain level. Based on the Parieto-Frontal Integration Theory (P-FIT)[6,19], we aimed to test hypotheses related to specificity of regional associations with intelligence. To this end, we utilized the multimodal parcellation scheme provided by the Human Connectome Project, which delineates 180 cortical brain regions per hemisphere[37]. NODDI coefficients from homotopic brain regions were averaged across both hemispheres, resulting in 180 mean values. The associations between these NODDI coefficients and intelligence were analyzed by means of partial correlations, controlling for age, sex, and all remaining cortical brain properties, while correcting for multiple comparisons using the Benjamini–Hochberg method (Fig. 3 and Supplementary Fig. 6).

For sample S259, the vast majority of brain regions exhibited negative associations between intelligence and $INVF_{Cortex}$ (159 out of 180 brain regions) as well as intelligence and $ODI_{Cortex}$ (164 out of 180 brain regions) (Fig. 3). However, none of the partial correlations involving $INVF_{Cortex}$ survived correction for multiple comparisons. In contrast, the negative associations between intelligence and $ODI_{Cortex}$ reached statistical significance in 12 brain regions with partial correlation coefficients in the range of $-0.21$ to $-0.18$. Importantly, the majority of these brain regions (9 out of 12) showed an overlap with brain regions from the original P-FIT model as defined by Jung and Haier[6] or its updated version proposed by Basten et al.[19] (see Methods).

Performing the same analysis for sample S498 resulted in 154 out of 180 brain regions showing negative associations between intelligence and $INVF_{Cortex}$ with partial correlation coefficients in 11 of these regions reaching statistical significance ($r = -0.19$ to $-0.14$) (Supplementary Fig. 6). Again, there was an overlap between the P-FIT model and some of the statistically significant brain regions (7 out of 11). Intelligence was negatively associated with $ODI_{Cortex}$ in 151 out of 180 brain regions. In five of these brain regions the respective partial correlations reached statistical significance with coefficients ranging from $-0.17$ to $-0.14$. Brain regions overlapping with the P-FIT model could be identified in four out of five cases.

## Discussion

The primary goal of this study was to investigate the relationship between intelligence and neuroanatomical correlates on both macroscopic and microscopic levels. To this end, we examined volume estimates of the whole-brain as well as single brain regions and utilized an advanced diffusion MRI technique to analyze the architecture of dendrites and axons.

Our data as well as data provided by the Human Connectome Project[25] revealed an expected positive association between cortical volume and intelligence, corrected for age, sex, and collinearity. It is a well-established and consistent observation that cognitive abilities are related to brain volume, especially the volume of the cerebral cortex[1,4,5]. The biological explanation for this structure–function relationship is usually derived from the fact that individuals with more cortical volume possess a higher number of neurons[7,8] and thus more computational power to engage in logic reasoning (Fig. 4). However, the major aim of our study was to investigate the microstructural architecture of the cortex by closely analyzing the diffusion characteristics of dendrites and axons.

We found that specific microstructural properties were associated with intelligence, especially in cortical regions included in the P-FIT network. Cortical gray matter is largely composed of the neuropil, namely, dendrites, axons, and glial cell processes. These structures restrict the movement of water molecules and are modeled as sticks in the NODDI model, from which markers, resembling neurite density and neurite orientation dispersion, can be computed[15–18]. Histological examinations have shown that the relative proportion of glial cells within a fixed volume of cortex is relatively small compared to other components[38,39]. The influence of their processes on the diffusion signal can thus be regarded as negligible. As a consequence, the diffusion signal arising from the intra-neurite space can be attributed to the architecture of dendrites and axons[15]. Our results indicate that neurite density and neurite orientation dispersion within the cortex are both negatively associated with intelligence. At first glance, this finding might appear counterintuitive to the central working hypothesis of differential neuroscience, which usually finds that "bigger is better" (i.e., more neuronal mass is associated with higher ability levels). However, our results conform well to findings on the mechanisms of maturation-induced and learning-induced synaptic plasticity. Brain maturation is associated with a sharp increase of synapse number, followed by a massive activity-dependent synaptic pruning that reduces synaptic density by half, thereby enabling the establishment of typical mature cortical microarchitecture[40]. Maturation-associated synaptic pruning is not only an event of early childhood, but proceeds at a rapid rate at least until the end of the second decade of life[41]. Most importantly, the mechanisms of synaptic growth and pruning

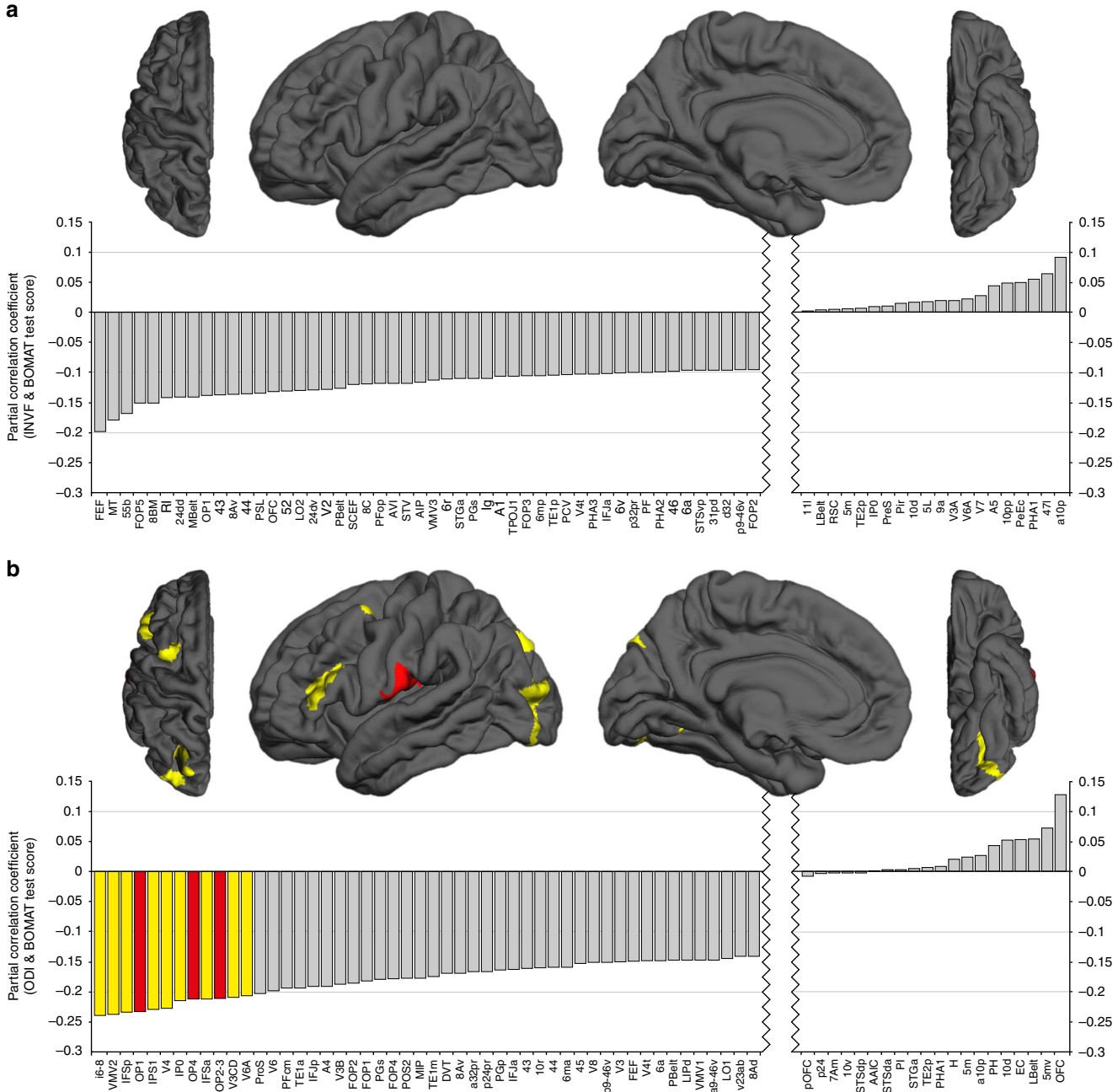

**Fig. 3** Partial correlation analyses with data from sample S259 quantifying structure–function associations at the level of single brain regions. For each hemisphere, 180 cortical brain regions were defined based on the multi-modal parcellation scheme provided by the Human Connectome Project. NODDI coefficients and volume measures from homotopic brain regions were averaged across both hemispheres, resulting in 180 mean values. Structure–function associations between $INVF_{Cortex}$ and intelligence (**a**) as well as $ODI_{Cortex}$ and intelligence (**b**) were analyzed by means of partial correlations with age, sex, and the remaining cortical brain properties as controlling variables. FDR correction using the Benjamini–Hochberg method was applied to account for a total of 180 comparisons. Partial correlation coefficients are depicted as gray bars arranged by magnitude from negative to positive. Due to space restrictions, a middle portion of 110 brain regions exhibiting no significant structure–function associations is spared out. Statistically significant partial correlation coefficients that survived a critical FDR threshold of $q = 0.05$ (see Methods) are highlighted in either red or yellow. The yellow color marks significant partial correlation coefficients that are exhibited by brain regions from the P-FIT model of intelligence. Following this color scheme, respective brain regions are marked in either red or yellow on a cortical surface. $INVF_{Cortex}$ = intra-neurite volume fraction representing neurite density in the cortex, $ODI_{Cortex}$ = orientation dispersion index of neurites in the cortex

during maturation overlap with those of learning in the mature brain[42]. Consequently, diverse learning tasks are associated with simultaneous growth and retraction of dendritic and synaptic processes in involved neural zones[43,44]. Microstructural studies with confocal imaging on organotypic brain cultures reveal that long-term potentiation initially induces synaptic growth, followed by an increased loss of connections within 10% of non-stimulated hippocampal spines[45]. Thus, both the ability to produce and prune neural connections constitutes the neurobiological foundation of learning and cognition.

Perturbations of synaptic and dendritic growth and pruning have grave consequences with regard to cognitive performance[46].

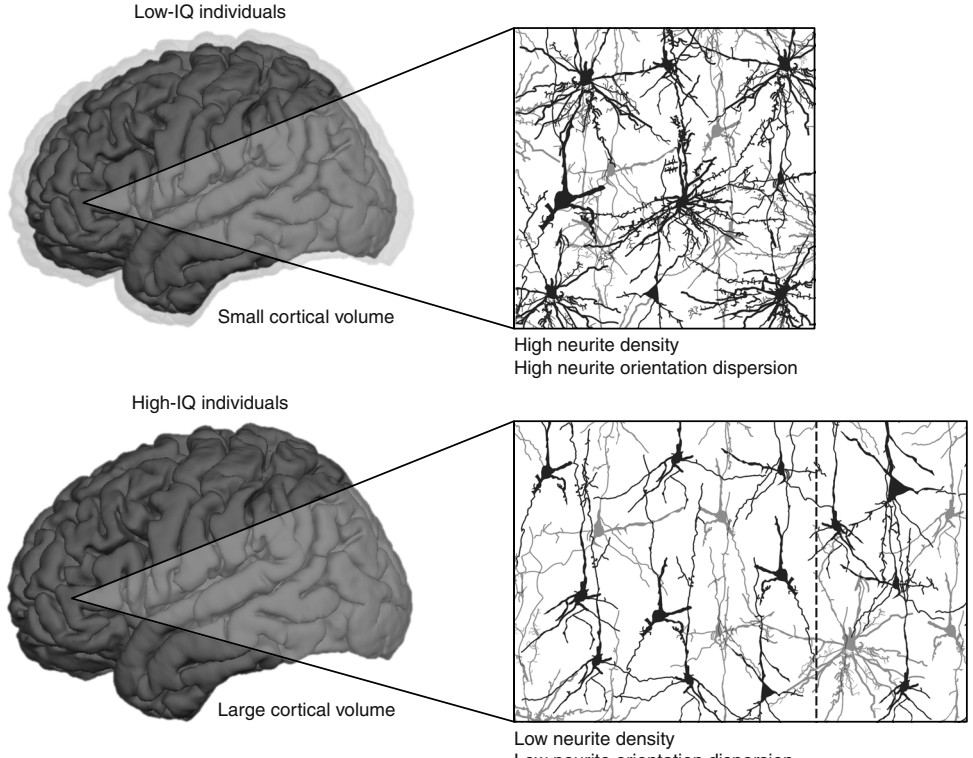

**Fig. 4** Schematic depiction of differences between low-IQ and high-IQ individuals with regard to brain volume, neurite density, and arborization of dendritic trees within the cortex. High-IQ individuals are likely to possess more cortical volume than low-IQ individuals, which is indicated by differently sized brains (left side) and differently sized panels showing exemplary magnifications of neuron and neurite microstructure (right side). The difference in cortical volume is highlighted by the shadow around the upper brain. Due to their larger cortices, it is conceivable that high-IQ individuals benefit from the processing power of additional neurons, which are marked by the dotted line in the lower panel. The cerebral cortex of high-IQ individuals is characterized by a low degree of neurite density and orientation dispersion, which is indicated by smaller and less ramified dendritic trees in the respective panel. Intellectual performance is likely to benefit from this kind of microstructural architecture since restricting synaptic connections to an efficient minimum facilitates the differentiation of signals from noise while saving network and energy resources. Neurons and neurites are depicted in black and gray to create a sense of depth. Please note, this depiction does not correspond to the actual magnitude of effect sizes reported in the study. For the purpose of an easier visual understanding, differences in both macrostructural and microstructural brain properties are highly accentuated

For example, reduced synaptic pruning results in an excess of synapses, which is associated with pathologies characterized by low intelligence including Down's syndrome[47,48]. An increase in synapses may also cause failure in differentiating signals from noise, reducing network efficiency[49]. Indeed, computational studies show that synaptic pruning increases learning and processing speed, and saves network and energy resources[50], by requiring less computation to learn relations between data sets[51]. These observations are in line with the results obtained from both our experimental data and validation data from the Human Connectome Project[25]. We found that both $INVF_{Cortex}$ and $ODI_{Cortex}$, representing neurite density and orientation dispersion in the cerebral cortex, were negatively associated with intelligence. Since both markers are closely related to the amount of synaptic connections, our findings provide the first evidence of specific microstructural brain correlates facilitating efficient information processing as measured by intelligence (Fig. 4). This supports the neural efficiency hypothesis of intelligence[10–12]. In the original PET study of neural efficiency[52], researchers examined two samples of low-IQ individuals, including patients suffering from Down's syndrome and another form of mental retardation, as well as a control group of individuals with average intelligence. They found that both low-IQ groups exhibited higher rates of cortical glucose metabolism compared to the healthy control participants while working on Raven's Advanced Progressive

Matrices[9,53]. They attributed their observations to a failure of neural pruning in the brains of low-IQ individuals[13,52]. It is very important to note that these researchers were restricted to a pathological sample when proposing a biological foundation for the neural efficiency hypothesis of intelligence. Given the lack of suitable post mortem data or practical in vivo methods to obtain information about cortical microstructure, they examined individuals that were known to have dendritic trees with a very distinct microstructure, i.e., patients with Down's syndrome. However, evidence from a clinical sample is prone to influence by various confounding factors. Therefore, one should proceed with utmost care when generalizing these findings to our results, which were obtained from healthy individuals in the range of average intelligence.

Nevertheless, there is some evidence from healthy subjects to support the idea that interindividual differences in intelligence are associated with different levels of cortical activation during reasoning. For example, early EEG studies showed that high-IQ individuals, when working on an elementary cognitive task, display an event-related desynchronization (ERD) limited to cortical areas required for the task[54]. In contrast, low-IQ individuals were characterized by an ERD that was spread across a wide range of cortical areas. We hypothesize that this evidence of unfocused cortical activity was associated with redundant neuronal circuits in the form of expendable dendrites in the cortex. In another EEG

study by Walhovd et al.[30] the authors demonstrated that the latency of the ERP component P3a, as a measure of speed-of-processing, was negatively correlated with intelligence. Again, these findings can be interpreted in terms of neural efficiency and correspond to the results presented in our study. Future studies utilizing both structural and functional techniques will be critical in determining whether a higher degree of neurite density and orientation dispersion could slow cortical speed-of-processing due to inefficient circuitry, thus having a negative effect on intelligence.

Taken together, the results of the present study contribute to our understanding of human intelligence differences in two ways. First, our findings confirm an important observation from previous research, namely, that bigger brains with a higher number of neurons are associated with higher intelligence. Second, we demonstrate that higher intelligence is associated with cortical mantles with sparsely and well-organized dendritic arbor, thereby increasing processing speed and network efficiency. Importantly, the findings obtained from our experimental sample were confirmed by the analysis of an independent validation sample from the Human Connectome Project[25]. This replication of results is particularly striking given that both data sets are very different on many levels. For example, two different cognitive tests were used in order to measure intelligence, i.e., BOMAT and PMAT24. Both of them are culture-fair matrix-reasoning instruments capable of assessing the construct of fluid intelligence. Nevertheless, both tests tend to produce different results when testing individuals from high-IQ ranges. This might be attributed to the fact that BOMAT, in contrast to PMAT24 and other matrix-reasoning tests, was deliberately designed to avoid ceiling effects in very intelligent samples such as university students or high potentials. Moreover, both data sets used for this study differ with regard to their MRI data. Although the diffusion-weighted data from sample S259 is of sufficient quality and meets current standards in the field of neuroscience, it goes without saying that the data provided by the Human Connectome Project is of higher quality in terms of data acquisition and preprocessing. For example, diffusion-weighted data from sample S498 is superior to sample S259 in terms of voxel size ($1.25 \times 1.25 \times 1.25$ mm vs. $2 \times 2 \times 2$ mm) and number of total diffusion directions (288 vs. 128). In addition to that, there are differences in the preprocessing protocols of both data sets as well. While the eddy_correct pipeline from FSL was used to correct for eddy-current-induced distortions in the S259 sample, the Human Connectome Project utilized FSL's recently published eddy tool for this task[55]. Another important aspect worth mentioning is that the two samples themselves are not completely equal to one another. The S259 sample includes 259 participants with about 53% of them being male, whereas the S498 sample features almost twice as much participants of which merely 41% are males. In view of all these differences, it is hardly surprising that there are some results from the S259 sample that do not exactly match those obtained from the Human Connectome Project's data. Nevertheless, we feel that the similarities far outweigh the minor differences. Both data sets indicate that intelligence is associated with neurite density and orientation dispersion. Equally important, both data sets also show that this association points into a negative direction. This general pattern is clearly visible in both data sets. Moreover, one has to acknowledge that most of the statistically significant cortical areas, despite lacking a perfect match between data sets, show an impressive overlap with regions previously identified as belonging to the P-FIT network (about 70%). Finally, to the best of our knowledge, these results are the first to offer a neuroanatomical explanation underlying the neural efficiency hypothesis of intelligence.

In conclusion, the results obtained by NODDI substantially extend our knowledge about the biological basis of human intelligence differences, by providing insight regarding the biological basis of efficiency of processing at the neuronal level. The complementary findings at both macrostructural and microstructural levels provide a comprehensive biological mechanism, adding to the growing body of literature supporting a distributed network of efficiently organized neurons and axons underlying the expression of human intelligence.

## Methods

**Participants in the S259 sample.** Two hundred fifty-nine participants (138 males) between 18 and 40 years of age (M = 24.31, SD = 4.41) took part in the study. Since this is the first study to investigate the microstructural correlates of intelligence using NODDI, it was not possible to estimate the necessary sample size a priori based on existing literature. Instead, we collected data from a reasonably large sample and computed the achieved power post hoc using G*Power[56]. The analysis was based on the multiple regression model reported for sample S259 (Table 1) ($f^2 = 0.16$, $\alpha = 0.05$, 10 predictors, 259 participants) and yielded a power of 0.99, thereby indicating sufficient sample size. Two hundred thirty-five participants were right-handed and the remaining 24 participants were left-handed as measured by the Edinburgh Handedness Inventory[57]. This ratio is representative of the human population[58]. All participants had normal or corrected-to-normal vision and hearing. They were either paid for their participation or received course credit. All participants were naive to the purpose of the study and had no former experience with the administered intelligence test. Participants had no history of psychiatric or neurological disorders and matched the standard inclusion criteria for fMRI examinations. Each participant completed the matrix-reasoning test and neuroimaging measurement described below. All behavioral and neuroimaging variables used for analyses on the whole-brain level were normally distributed according to a Kolmogorov–Smirnov test. All data were checked for extreme outliers as defined by Tukey's fences[59] (observations three interquartile ranges below the first or above the third quartile, respectively), but none were found. Thus, no observations were excluded. The study was approved by the local ethics committee of the Faculty of Psychology at Ruhr-University Bochum. All participants gave their written informed consent and were treated in accordance with the declaration of Helsinki.

**Participants in the S498 sample.** For the purpose of validating the results obtained from sample S259, recruited at Ruhr-University Bochum, we downloaded additional data provided by the Human Connectome Project, namely, the "S500 plus MEG2" release[25]. This set includes 506 participants with data suitable for our analyses. We excluded eight participants because of extreme outliers being detected in their behavioral or neuroimaging data[59]. Thus, all of the reported analyses were performed on data from 498 participants (202 males) between 22 and 36 years of age (M = 29.16, SD = 3.48). Again, we performed a post hoc analysis using G*Power[56] in order to compute the achieved power. Based on the multiple regression model reported for sample S498 (Supplementary Table 4) ($f^2 = 0.09$, $\alpha = 0.05$, 10 predictors, 498 participants), the analysis resulted in a power of 0.99 and indicated sufficient sample size. As with sample S259, all neuroimaging variables used for analyses on the whole-brain level were normally distributed according to a Kolmogorov–Smirnov test. The PMAT24 test scores did not follow a normal distribution but were slightly skewed to the left. For the sake of comparability, sample S498 was analyzed in the same way as sample S259.

**Acquisition of behavioral data in the S259 sample.** The acquisition of behavioral data was conducted in a group setting of up to six participants, seated at individual tables, in a quiet and well-lit room. Intelligence was measured with a German matrix-reasoning test called BOMAT[21], which is widely used in neuroscientific research[60–62]. The test examines non-verbal mental abilities that contribute to intelligence and is similar to Raven's Advanced Progressive Matrices[9]. We conducted the "advanced short version" of the BOMAT, which has the advantage of high discriminatory power in samples with generally high intellectual abilities, thus avoiding possible ceiling effects[60]. The BOMAT inventory comprises two parallel test forms (A and B) with 29 matrix-reasoning items each. Participants had to complete only one of the two test forms, which were randomly assigned. Split-half reliability of the BOMAT is 0.89, Cronbach's $\alpha$ is 0.92, and parallel-forms reliability between A and B is 0.86[21]. Additionally, convergent and predictive validity are given for both BOMAT test forms since they are strongly correlated with other intelligence inventories ($r = 0.59$), tests of perceptual speed ($r = 0.51$), and German high school GPA ($r = -0.35$)[21]. The recent norming sample consists of about 2100 individuals with an age range between 18–60 years and equal sex representation.

**Acquisition of behavioral data in the S498 sample.** As with sample S259, intelligence was measured with a matrix-reasoning test, namely, the Penn Matrix Analysis Test (PMAT24)[26]. This instrument is included in the Computerized

Neuropsychological Test Battery provided by the University of Pennsylvania (PennCNP). The PMAT24 is an abbreviated version of the Raven's Progressive Matrices and includes 24 items of increasing difficulty. Each matrix pattern is made up of $2 \times 2$, $3 \times 3$, or $1 \times 5$ arrangements of squares with one of the squares missing. The participant must pick one of five response choices that best fits the missing square on the pattern. There is no time limit to the completion of the test, although the task discontinues if the participant makes five incorrect responses in a row. The PMAT24 has two test forms of which the Human Connectome Project only used one (form A) in order to assess intelligence.

**Acquisition of imaging data in the S259 sample.** All imaging data were acquired at the Bergmannsheil hospital in Bochum (Germany) using a Philips 3T Achieva scanner with a 32-channel head coil.

For the purpose of segmenting brain scans into gray and white matter segments as well as for the identification of anatomical landmarks, a T1-weighted high-resolution anatomical image was acquired (MP-RAGE, TR = 8179 ms, TE = 3.7 ms, flip angle = 8°, 220 slices, matrix size = 240 × 240, voxel size = 1 × 1 × 1 mm). The acquisition time of the anatomical image was 6 min.

For the analysis of NODDI coefficients, diffusion-weighted images were acquired using echo planar imaging (TR = 7652 ms, TE = 87 ms, flip angle = 90°, 60 slices, matrix size = 112 × 112, voxel size = 2 × 2 × 2 mm). Diffusion weighting was based on a multi-shell, high-angular-resolution scheme consisting of diffusion-weighted images for *b*-values of 1000, 1800, and 2500 s/mm², respectively, applied along 20, 40, and 60 uniformly distributed directions. All diffusion directions within and between shells were generated orthogonal to each other using the MASSIVE toolbox[63]. Additionally, eight data sets with no diffusion weighting (*b* = 0 s/mm²) were acquired as an anatomical reference for motion correction and computation of NODDI coefficients. The acquisition time of the diffusion-weighted images was 18 min.

**Acquisition of imaging data in the S498 sample.** All imaging data included in sample S498 were acquired on a customized Siemens 3T Connectome Skyra scanner housed at Washington University in St. Louis using a standard 32-channel Siemens receive head coil. Anatomical and diffusion-weighted imaging were carried out on two separate days with a mock scanner practice preceding the anatomical imaging on the first day. The Human Connectome Project's imaging hardware and protocols are documented elaborately in the reference manual for the "S500 plus MEG2" release.

A T1-weighted high-resolution anatomical image was acquired by means of an MP-RAGE sequence and the following parameters: TR = 2400 ms, TE = 2.14 ms, flip angle = 8°, matrix size = 224 × 224, voxel size = 0.7 × 0.7 × 0.7 mm. The acquisition time of the anatomical image was 7 min and 40 s.

The Human Connectome Project provides diffusion-weighted data suitable for the analysis of NODDI coefficients. The respective images were acquired using echo planar imaging and the following parameters: TR = 5520 ms, TE = 89.5 ms, flip angle = 78°, 111 slices, matrix size = 168 × 144, voxel size = 1.25 × 1.25 × 1.25 mm. The diffusion-weighted imaging session included six runs based on three different gradient tables once acquired in the right-left and left-right phase-encoding directions. The gradient tables included 90 diffusion weighting directions and six acquisitions with *b* = 0 s/mm² interspersed throughout each run. As with the data obtained for sample S259, diffusion weighting consisted of three shells, in this case *b* = 1000, 2000, and 3000 s/mm² interspersed with an approximately equal number of acquisitions on each shell within each run. Each of the six runs lasted approximately 9 min and 50 s, thereby, overall acquisition time amounted to about an hour.

**Analysis of imaging data in the S259 sample.** We used published surface-based methods in FreeSurfer (http://surfer.nmr.mgh.harvard.edu, version 5.3.0) to reconstruct the cortical surfaces of the T1-weighted images. The details of this procedure have been described elsewhere[22,23]. The automated reconstruction steps included skull stripping, gray and white matter segmentation, as well as reconstruction and inflation of the cortical surface. After preprocessing, each individual segmentation was quality controlled slice by slice and inaccuracies for the automated steps were corrected by manual editing if necessary. The automated brain segmentation yielded an estimate of the overall cortical volume (VOL$_{Cortex}$) and the overall white matter volume (VOL$_{WM}$). For the purpose of analyzing our data with regard to structure–function relationships on the level of single brain regions, we utilized the Human Connectome Project's multi-modal parcellation (HCPMMP)[37]. This parcellation scheme delineates 180 cortical brain regions per hemisphere and is based on the cortical architecture, function, connectivity, and topography from 210 healthy individuals. The original data provided by the HCP were converted to annotation files matching the standard cortical surface in FreeSurfer called fsaverage. This fsaverage parcellation was transformed to each participant's individual cortical surface and converted to volumetric masks. In a final step, the two segments delineating the overall cortex and white matter as well as the 360 masks representing single cortical brain regions yielded by the HCPMMP were linearly transformed into the native space of the diffusion-weighted images (Fig. 1, left box). The transformed regions served as anatomical landmarks from which NODDI coefficients were extracted (Fig. 1, right box).

Diffusion images were preprocessed using FDT (FMRIB's Diffusion Toolbox) as implemented in FSL version 5.0.7. Preprocessing steps included a correction for eddy currents and head motion using the eddy_correct tool. Subsequently, gradient directions were corrected to account for any reorientations in the eddy_correct output. NODDI coefficients were computed using the AMICO toolbox[24]. The AMICO approach is based on a convex optimization procedure that converts the non-linear fitting into a linear optimization problem[24]. This reduces processing time dramatically[64]. Data analysis with NODDI can be applied to cortical regions as well as white matter structures. However, it is necessary to optimize the NODDI model for the purpose of analyzing gray matter structures since different types of brain tissue may vary considerably with regard to their intrinsic free diffusivity[18,65,66]. Because of this, we adjusted the AMICO toolbox and changed its respective parameter for intrinsic free diffusivity to $1.1 \times 10^{-3}$ mm²/s for analyzing gray matter structures and utilized the toolbox' default setting of $1.7 \times 10^{-3}$ mm²/s for the analysis of white matter. The NODDI technique is based on a two-level approach and features a three-compartment model distinguishing intra-neurite, extra-neurite, and CSF environments. First, the diffusion signal obtained by the multi-shell high-angular-resolution imaging protocol is used to determine the proportion of free moving water within each voxel[15–17,24,67]. This ratio is termed isotropic volume fraction and reflects the amount of isotropic diffusion with gaussian properties likely to be found in the CSF of gray (ISO$_{Cortex}$) and white matter (ISO$_{WM}$) regions. Second, the remaining portion of the diffusion signal is attributed to either intra-neurite environments or extra-neurite environments[15–17]. The proportion of intra-neurite environments is quantified as the intra-neurite volume fraction (INVF). INVF represents the amount of stick-like or cylindrically symmetric diffusion that is created when water molecules are restricted by the membranes of neurites. In white matter structures this kind of diffusion (INVF$_{WM}$) is likely to resemble the proportion of axons. In gray matter regions (INVF$_{Cortex}$) it serves as an indicator of dendrites and axons forming the neuropil. Extra-neurite environments are characterized by hindered diffusion and are usually occupied by various types of glial cells in white matter structures and both neurons and glial cells in gray matter regions[15–17].

Neurite orientation dispersion is a tortuosity measure coupling the intra-neurite space and the extra-neurite space, resulting in alignment or dispersion of axons in white matter (ODI$_{WM}$) or axons and dendrites in gray matter (ODI$_{Cortex}$)[17,67]. Examples of INVF, ODI, and ISO coefficient maps from a representative individual are illustrated in Fig. 1, right box. As described above, the cortical and white matter regions defined for the T1-weighted anatomical scans were transformed into the native space of the diffusion-weighted images to compute NODDI coefficients for areas across the whole brain.

**Analysis of imaging data in the S498 sample.** The analyses of anatomical and diffusion-weighted data from sample S498 were carried out in the same way as described for sample S259. The only differences in analyses were found in their preprocessing. For example, the Human Connectome Project utilizes a combination of the FSL tools topup and eddy in order to correct for eddy currents, head motion, and EPI distortions simultaneously. These tools represent an updated version of the eddy_correct tool used for the S259 sample and make use of the fact that one-half of the HCP's diffusion-weighted data was acquired in the right-left phase-encoding direction and the other half in the left-right phase-encoding direction. The HCP's preprocessing pipelines for anatomical and diffusion-weighted data are detailed in the reference manual for the "S500 plus MEG2" release as well as in Glasser, Sotiropoulos[68].

**Matching single brain regions against the P-FIT model.** Subsequent to the analysis of structure–function relationships on the level of single brain regions, all brain regions showing statistically significant associations between NODDI coefficients and intelligence were matched against the P-FIT model[6,19]. To this end we employed a cortical parcellation based on Brodmann areas[69], which is included as annotation files named "lh.PALS_B12_Brodmann" and "rh.PALS_B12_-Brodmann" in FreeSurfer. By using FreeSurfer's aparc2aseg tool, both files were converted to a volumetric segmentation matching the cortex of the fsaverage standard brain. The same was done to the HCPMMP annotation file. By means of an in-house Matlab program, each brain region included in the HCPMMP was assigned to one of the Brodmann areas. This was done by comparing each voxel within a HCPMMP region to its corresponding voxel from the Brodmann segmentation. The Brodmann area showing the largest overlap with the respective HCPMMP region was identified in terms of number of matching voxels. In the original version proposed by Jung and Haier[6], the P-FIT features a network of 14 Brodmann areas. In an updated version by Basten et al.[19] the network's composition was confirmed, but also extended to five additional Brodmann areas. If our partial correlation analyses yielded a statistically significant brain region that was assigned to one of these 19 Brodmann areas, it was considered to belong to the P-FIT model.

**Statistical analysis.** Statistical analyses were carried out using Matlab, version 7.14.0.739 (R2012a, The MathWorks Inc., Natick, MA) and SPSS version 20 (SPSS Inc., Chicago, IL). For all analyses, linear parametric methods were used. Testing was two-tailed with an α-level of 0.05, which was FDR corrected for multiple

comparisons using the Benjamini–Hochberg method[70] when conducting correlation analyses on the level of single brain regions.

We examined structure–function relationships on a whole-brain level by computing partial correlation coefficients reflecting the associations between intelligence and the structural brain properties included in this study. Age and sex were used as controlling variables. We followed a similar but more stringent approach for our analyses on the level of single brain regions. As described above, the parcellation scheme provided by the Human Connectome Project[37] yielded 180 cortical regions per hemisphere. NODDI coefficients and volume measures from homotopic regions were averaged across both hemispheres, resulting in 180 mean values for $INVF_{Cortex}$, $ODI_{Cortex}$, $ISO_{Cortex}$, and $VOL_{Cortex}$, respectively. The associations between $INVF_{Cortex}$ and intelligence as well as $ODI_{Cortex}$ and intelligence were analyzed by means of partial correlations, controlling for age and sex, and the remaining cortical brain properties, while correcting for multiple comparisons using the Benjamini–Hochberg method[70].

To examine the structure–function relationships with regard to the unique contribution of each brain property included in the correlation analyses, we computed a multiple regression analysis using SPSS. Intelligence was treated as the dependent variable and $INVF_{Cortex}$, $INVF_{WM}$, $ODI_{Cortex}$, $ODI_{WM}$, $ISO_{Cortex}$, $ISO_{WM}$, $VOL_{Cortex}$, $VOL_{WM}$, age, and sex as predictors.

**Code availability**. The Matlab code that was used to compute the overlap between statistically significant brain regions and those included in the P-FIT model is available from the corresponding author upon reasonable request.

**Data availability**. The data that support the findings of this study are available from the corresponding author upon reasonable request. The data used for sample S498 are part of the "S500 plus MEG2" release provided by the Human Connectome Project and can be accessed via its ConnectomeDB platform (https://db.humanconnectome.org/).

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

## Acknowledgements

This work was supported by the Deutsche Forschungsgemeinschaft (DFG) grant numbers Gu227/16-1, GE2777/2-1, and DFG SFB 1280 project A03 and the MERCUR foundation grant number An-2015-0044. The authors thank Lara Schlaffke, Martijn Froeling, and PHILIPS Germany (Burkhard Mädler) for their scientific support with the MRI measurements, as well as Tobias Otto for his technical support. Data were provided in part by the Human Connectome Project, WU-Minn Consortium (Principal Investigators: David Van Essen and Kamil Ugurbil; 1U54MH091657) funded by the 16 NIH Institutes and Centers that support the NIH Blueprint for Neuroscience Research; and by the McDonnell Center for Systems Neuroscience at Washington University.

## Author contributions

E.G. conceived the project and supervised the experiments. E.G., C.F., O.G., and R.J. designed the experiments. C.F., C.S., and P.F. performed the experiments. E.G., C.F., R.H., M.V., and J.L. analyzed the data. E.G., C.F., and R.J. wrote the manuscript and all authors edited the manuscript.

## Additional information

**Competing interests:** The authors declare no competing interests.

