## [Peer Review File · Nature Communications]

Reviewers' comments:

Reviewer #1 (Remarks to the Author):

The authors present a very interesting investigation about the relationship between fluid intelligence and intrinsic grey/white matter properties of the brain. The work leverage prior evidence by Richard Haier, who, by means of PET and fMRI studies on intelligence, has put forward the idea of intelligence being more related to computational/processing efficiency than the mere number of neurons in each individual brain. As the authors discuss in the Introduction section of the manuscript, several attempts to validate Haier's "neural efficiency" theory have been made in the last three decades, but no convincing argument is available so far.

The approach used in the present investigation might represent a suitable framework to test such hypothesis, and the authors carefully designed their study by including a large sample of healthy participants and appropriate neuroimaging sequences able to capture brain volume, neuron density and arborization of dendritic trees. However, apart from a few statistical issues listed below, the study suffers from one crucial issue related to the behavioral assessment of fluid intelligence. See my comments below, I hope they will help improve this manuscript as well as suggest some alternative options for gf measurement in future studies.

General comment

1) The introduction is well written and provide the readers with a very comprehensive overview of the issue at hand, also adequately conveying the message about the importance of the present investigation for the understanding of the neurobiological underpinnings of intelligence in humans. I also think the method applied in the present investigation really represents the right tool to answer the Neural Efficiency hypothesis. Congratulations on the insight behind the study design.

2) As previously mentioned, they study has the potential to address a crucial issue with current theories about the neurobiological underpinnings of fluid intelligence in humans. Therefore, it is extremely important that the results do not suffer from any potential bias, especially when it comes to the definition and measurement of the cognitive construct/function of interest. Given the potential impact of the present study, a biased observation might lead to a sequelae of replication studies, as well as others trying to expand such evidence in other directions. This might lead to misleading conclusions and attempts to replicate the present results, which in my opinion are not strictly related to "fluid intelligence". Specifically, the authors used the BOMAT task, a recently developed matrix-reasoning task thought of measuring gf in humans. Unfortunately, compared to the gold-standard approach for gf assessment, i.e. the matrix reasoning task by Raven (Raven J., Raven J.C., & Court J.H., 1998), the BOMAT suffers from several issues which had brought many researchers to label it more as a working memory task than a pure fluid intelligence one. For instance, Moody clearly address this issue (Moody, 2009) by highlighting how the test, which includes a different parameter space for alternative solutions (i.e. the BOMAT items have a 5*3 grid of alternative solutions = 15) compared to

any other matrix-like task, including the Raven Progressive Matrix (RPM, 3*3) or the Sandia logical/relational test (3*3) (Matzen et al., 2010). Given the increased number of possible answers available to the subject, the BOMAT stresses visuo-spatial working memory significantly more than RPM or Sandia, by reducing the pure contribution of logical reasoning in favor of the ability of holding and manipulating items in each subject's working memory storage. In my personal experience, and please consider it as just an additional evidence, the BOMAT does not measure exactly the same construct as RPM and other matrix reasoning tasks: we have recently analyzed data on ~200 participants who underwent both RPM, Sandia and BOMAT gf assessment, as well as structural and functional MRI acquisitions before and after an executive function training aimed at increasing gf. Factorial analysis has shown how the BOMAT load on a different component with respect to Sandia and RPM, and not surprisingly it shows significant gains after training while neither RPM nor Sandia do.

Unfortunately, the selection of gf measures represents a crucial element to push forward a very relevant argument as human intelligence being more related to one or another brain property. If the study gets published and other groups will not be able to replicate the finding by using, for instance, the RPM, it will be difficult to re-label the present findings. To have a first evidence with a more validated measure, or possibly a latent factor level gf score obtained by using multiple gf measures, would be ideal.

Specifically, why did the author decide to use the BOMAT instead of the Raven's matrix or other more similar tasks? Do they have other gf tasks that could be used to validate current results? I assume they do not, but if I am wrong I would suggest running the same analysis with the RPM.

Matzen, L. E., Benz, Z. O., Dixon, K. R., Posey, J., Kroger, J. K., & Speed, A. E. (2010). Recreating Raven's: software for systematically generating large numbers of Raven-like matrix problems with normed properties. *Behav.Res.Methods*, 42(1554–3528 (Electronic)), 525–541. <https://doi.org/10.3758/BRM.42.2.525>

Moody, D. E. (2009). Can intelligence be increased by training on a task of working memory? *Intelligence*, 37(4), 327–328. <https://doi.org/10.1016/j.intell.2009.04.005>

Raven J., Raven J.C., & Court J.H. (1998). Manual for Raven's progressive matrices and vocabulary scales.

3) Overall, the manuscript suggests a strong association between Noddi derived brain properties and fluid intelligence, however the correlation coefficients reported in the manuscript are relatively small ($r = -.19/21$, which mean that only ~4% of variance in intelligence is explained by the different brain features included in the analysis). This is also relevant when the argument about BOMAT is considered: given the very little variance explained by Noddi-indexes, and the potential "gf measurement error", the overall

correlation pattern might be actually reflecting noise in the data as well as a relationship between brain properties and an unspecific cognitive construct more related to speed of processing (which constitute a relevant component of BOMAT problem solving, given the higher number of alternative solutions to be tested simultaneously under time pressure) or, as previously mentioned, working memory capacity/processing.

MINOR comments

1) Are the correlation results corrected for multiple comparisons?

2) In the Result section, the author mention a cortical parcellation they used to look at regional correlation between gf and Noddi measures. Neither information about the nature of the parcellation (structurally or functionally defined regions?) nor the total number of regions (authors mention "34 regions" while the atlas includes 34 regions per hemisphere = 68, see Figure 1) are provided. I would suggest including these information, as well as a rationale for selecting this atlas over many other available ones. Also, I would include such details in the Methods section.

3) Results section: "Intelligence was significantly associated with ODIcortex in 9 out of 34 brain regions ($r = -.17$ to $-.23$) with most of the identified regions being part of the Parieto-Frontal Integration Theory (P-FIT) of intelligence 7 (Fig. 4A)". Are the regions included in the analysis 34 or 68 (see previous comment)? If the latter (as suggested in Figure 1), are the significant regions 9 per hemisphere or 9 in total? Moreover (see below)

4) the authors also suggest that "...most of the identified regions being part of the Parieto-Frontal Integration Theory (P-FIT) of intelligence". How has this been quantified? The atlas used by the authors is the Desikan-Killiany, which has not been used in the original definition of the P-FIT; given the difference among available cortical and subcortical parcellations, it could be that a region labelled as Superior parietal lobe in the present work does not exactly correspond to the same region in the original P-FIT model. This is particularly true for the anatomical parcellation of regions in the prefrontal and parietal lobes where, for instance, atlas like the AAL, Harvard-Oxford and FSL offer completely different parcellation schemes. Did the author computed a Dice coefficient with a P-FIT template?

5) Discussion section: "These results indicate that all of the three brain properties linked to the cortex contribute to the prediction of intelligence independently of each other. In contrast to the results yielded by the correlation analysis, intelligence was no longer significantly associated with ODIWM. Further, we observed no significant associations between intelligence and the remaining predictors INVFWM, VOLWM, age and sex.". It seems the authors attribute the difference in results to the application of correlation vs regression analyses. Did they try running partial correlations? Results might be similar to those obtained using regression analysis, offering the reader a much more linear sequence of analyses to follow.

6) Some comments on Autism, brain properties and intelligence in the Discussion section could be toned down. They are relevant to the topic and findings, but given the lack of an equivalent analysis in this patient population using Noddi, I would suggest to limit this reference to a sentence instead of a whole paragraph.

7) The same applies to the sentence "This hypothesis is also backed by a PET study in a sample of low-IQ individuals, including patients suffering from Down's syndrome and another form of mental retardation 50. Both low-IQ groups exhibited higher rates of cortical glucose metabolism compared to healthy control participants while working on Raven's Advanced Progressive Matrices. This observation was attributed to a failure of neural pruning in the brains of low-IQ individuals. It is important to note that we worked with mentally healthy subjects and not with a pathological sample. However, there is evidence that the same mechanisms discussed above could also apply to healthy individual with individually different IQs, albeit on a smaller scale.". This is not true and the present data does not support this kind of statement. The authors should not compare a significant but very weak evidence (correlation ~ 0.2) in healthy participants with clinical conditions characterized by pathological genetic and developmental substrates. Given this represents the first evidence of a neurite density correlation with intelligence, I would exclude any reference to pathological conditions in the manuscript.

8) Figure 5 is highly speculative. Even though this reflect its main purpose (that is providing the reader with a simplified interpretation of the findings) the nature of the publication should reinforce a more technical depiction of the results. Most importantly, the figure itself suggest a striking –visible to the naked eye– difference between high and low IQ subjects in terms of brain volume, neuron density and arborization of dendritic trees. Instead, the magnitude of the correlations found in the study (e.g. 0.19) is very low (probably corresponding to an effect size of 0.1 or less). I would encourage the authors to carefully edit their manuscript, toning down the overall message and aiming at a more technical report.

Reviewer #2 (Remarks to the Author):

This study aimed to address the fundamental question of what are the biological substrates of intelligence. Previous research has shown that on one hand gray matter volume correlates positively with intelligence, on the other intelligent individuals show reduced brain activity during reasoning. But little is known about the brain tissue microstructure that underlies these observations. This study collected the data that may fill this gap.

The central findings of this study, if correct, will clearly represent a major step forward in our understanding of the biological origins of intelligence. In addition to replicating the previous finding that gray matter volume correlates positively with intelligence, the study also concludes that 1) there are more neurons per unit volume in higher intelligence individuals; 2) higher intelligence individuals exhibit lower neurite density and lower neurite

orientation dispersion. The latter point suggests that the higher intelligence individuals possess a sparser inter-neuron connective architecture. This may be interpreted as a more efficient architecture that in turn may explain the reduced brain activity seen in the higher intelligence individuals.

These findings were drawn based on the data acquired and analysed with a new MRI technique called NODDI. The technique is known to provide three independent markers of tissue microstructure: INVF, which is the neurite density, ODI, which is the neurite orientation dispersion, and ISO, which is the fraction of free water. Interestingly, the authors have chosen to analyze an additional parameter, namely, ENVF, which depends COMPLETELY on INVF, specifically, $ENVF = 1 - INVF$ by definition. They have chosen to analyze this parameter because they argue that it captures the neuron density. This interpretation is a significant departure from what was put forward by the developer of the NODDI technique, which has some unintended consequences: it implies that higher number of neurons (i.e. higher ENVF) can ONLY mean lower number of neurites (i.e. lower INVF), which is probably not warranted. With this in mind, the conclusion concerning the neuron density is questionable. Nevertheless, the conclusion regarding the neurite density and orientation dispersion provides sufficient novelty.

One minor critique concerns the inaccurate description of NODDI as being validated histologically. While there may well be new histological validation of NODDI emerging, the two studies cited (16 and 17) are the validation of an ex vivo MRI technique that the NODDI technique is based on; they are not the validation of NODDI themselves.

Reviewer #3 (Remarks to the Author):

The authors present interesting results of an inverse correlation between cortical neurite density and intelligence. In general, they use data of moderate quality and use methods that are similarly of moderate quality. There are several additional analyses and methodological improvements that should be easy for the authors to carry out, but should substantially increase the impact of the paper and ensure that confounds are not driving the results.

Major Points:

- A) The parcellation used in this study is the 34-region gyral and sulcal geographic parcellation of FreeSurfer. This parcellation has little to do with brain structure or function, as it is simply based on cortical folding patterns that are known to vary relative to cortical areas. The paper would have higher impact if it used the recently published multi-modal cortical parcellation from the Human Connectome Project. Methods have been released to map this parcellation to fsaverage, and from there the authors could map it to their individual subject surfaces and then to individual subject volumes and on to subject native diffusion space. The authors could then see if any of the cortical areas had particular relationships between NODDI metrics and intelligence. The use of a non-functionally relevant parcellation may be why few of the parcellated statistical tests proved significant.
- B) The diffusion data are of moderate spatial resolution (2x2x2mm), this is certainly

standard in the field and less than the mean cortical thickness of ~ 2.6 mm. That said, I wondered about the effects of partial voluming when measuring cortical properties. I also wondered about whether cortical thickness and surface area are also correlated with intelligence. Because the authors have run FreeSurfer on all their subjects, it should be trivial to add these parameters to their analyses. In particular I wonder if the multi-variate analysis would still show effects of these diffusion parameters after cortical thickness differences are accounted for. Also I wonder if these effects would be replicated using the much larger and higher resolution HCP (1.25mm isotropic) diffusion data where behavioral measures of intelligence are also available. I think all of these analyses would increase the impact of this study.

C) I also wondered about the effect of motion on these measures. Might more intelligent subjects also be more cooperative in the scanner, moving less relative to less intelligent subjects who might move more and have biases in their structural or diffusion data? I think a summary measure of motion (e.g. number of volumes above a frame-wise displacement (FD) threshold of X or one of the many other summary motion measures) should be included in the analyses as a covariate of no interest.

D) The authors don't really explain why ODI is negatively correlated with intelligence. One explanation is that ODI in the cortex is driven largely by partial volume effects (or motion might be another good cause) and thinner cortex would have more partial volume effects (and might also be associated with lower intelligence). This gets back to point B where it would be helpful to see if the ODI effect remains after partialling out cortical thickness variability.

E) It is necessary to estimate the intrinsic free diffusivity parameter for grey matter instead of just using the parameters for white matter and CSF e.g. using the procedure introduced by Guerrero et al. (2016) 'Investigating the effects of intrinsic diffusivity on neurite orientation dispersion and density imaging (NODDI)' (<http://dev.ismrm.org/2016/1046.html>). Was this done in this study?

Minor Points:

- 1) Awkward phrasing: Page 3 line 46 "has been met with" (in general the manuscript would benefit from further editing for typos and style)
- 2) Please give the range and standard deviation of the participants ages.
- 3) Please give the mean, range, and standard deviation of the intelligence scores being correlated with NODDI.
- 4) It is not necessary to call things "highly significant" just because they have p-values that are more than a little bit lower than 0.05. Perhaps such hyperbole should be reserved for those things that have very small p values requiring scientific notation to display.
- 5) Page 7 line 147: "designed an integrative neural prediction model" this seems like an unnecessarily complicated description of a basic (and appropriate) multi-variate analysis
- 6) Page 7 line 151: typo "included"
- 7) Page 9 line 181: awkward phrasing: "It was the overarching goal of this study" to "The overarching goal of this study was..."
- 8) Page 9 line 198: awkward phrasing: "bring forth stick-like diffusion patterns from which" to "are modeled as sticks in the NODDI model, from which NODDI markers resembling..."
- 9) There is no mention in the methods of how the diffusion data were registered to the T1w data, nor of how b0 distortions in the diffusion data were removed so that this registration

was accurate. A very accurate registration is necessary so that the cortical ribbon in the diffusion data lines up with the cortical ribbon of the structural data – even being off by a few mm will lead to a major mismatch as the cortex is only 2.6mm thick on average. Also it is not clear if eddy or eddy_correct was used to do motion and eddy current correction. Eddy is the preferred software tool for this.

Reviewer #4 (Remarks to the Author):

Review of NCOMMS-17-07793: Diffusion markers of dendritic density and arborization in gray matter predict differences in intelligence

In this study of 221 neurotypical individuals (18-35-years; 115 males), the authors investigate the relationship between diffusion-imaging markers of dendritic density and arborization in grey matter. The authors used diffusion-weighted magnetic resonance imaging (DW-MRI) and a specific diffusion reconstruction method called neurite orientation dispersion and density imaging (NODDI). The authors reported associations between neuron density/arborization correlates and intelligence as measured by a matrix reasoning test.

There are a number of strengths of this study. The question is interesting in general, the authors apply a multi-shell diffusion scheme on a large sample appropriate for using the NODDI reconstruction. That reconstruction algorithm is itself novel and innovative.

Methodologically the study is very sound. The multi-shell acquisition has sufficient directions (120) and shells (3) to do the NODDI reconstruction, and the resolution is good (2mm isotropic). Reconstruction steps were all appropriate and consistent with standards in the field. The use of a normed and reliable (split-half = .89) matrix reasoning test to assess intelligence is reasonable.

My major concern is whether the authors, in the correlation analysis, are p-hacking their way to significant effects, and those effects are very small. I don't mean this in a derogative way—it is important to explore the data. However, if the effects simply pop out of running a large number of correlations, that is another matter. For example, the main correlations with the whole brain measures have effect sizes of $r = .19$ and $r = .14$. The data from Figure 2 are not very convincing that there is a real effect in terms of the size, although both are significant. But would they replicate? It is not clear these were FDR corrected—the text is ambiguous. It says "Testing was two-tailed with an α -level of .05 which was FDR corrected for multiple comparisons when conducting correlation analyses on the level of single brain regions". This seems to imply that FDR correction was not applied to the whole-brain analysis, although the authors could clarify this, as I may be misreading.

The regression analysis is more convincing, and my opinion is that the authors should skip reporting the correlation analysis. Correcting for age and sex seems important. The authors could report on co-linearity, but they were careful to leave out ENFV so it is likely they examined this. In addition, the R-squared value should be reported. Is it explaining a lot of

variance? The advantage here is we see 1) the slightly larger effect sizes; 2) the fact that age and sex are not significant predictors.

An added variable plot for the whole brain analysis may also be more convincing, and this could replace Figure 2. I don't know if SPSS computes these, but, for example, R allows their computation (e.g., `avPlots` in R:

<https://www.rdocumentation.org/packages/car/versions/2.1-4/topics/avPlots>). In combination with Figure 3, this would show visually how well the data show the linear relation, controlling for other variables. If the regression is applied to Figure 4, the same presentation could be made (leaving out age and sex).

Finally, to try to avoid fishing for findings, the authors may want to compute the measures for brain regions that would be very unlikely to be associated with intelligence, and show that they are not. For example, would metrics in primary cortical regions be associated with intelligence? What about cerebellum?

Minor: The authors repeatedly write that larger brains = more neurons, but this is not necessarily the case because the type/size of neuron matters, and the density (e.g., there are more neurons in the cerebellum than the cortex). Perhaps they could correct this.

Anthony Dick

**Response to Reviewers:
Diffusion markers of dendritic density and arborization in gray matter
predict differences in intelligence**

Erhan Genç, Christoph Fraenz, Caroline Schlüter, Patrick Friedrich, Rüdiger Hossiep, Manuel C. Voelke, Josef M. Ling, Onur Güntürkün and Rex E. Jung

General Remarks

The four reviewers found this work of considerable interest and highlighted its importance for understanding the neurobiological underpinnings of intelligence in humans. We thank the reviewers for their valuable comments, which have led to a substantial improvement of the manuscript. We hope that our response, especially the additional analysis of data from the Human Connectome Project, addresses the issues raised by the reviewers in a satisfactory manner and that the manuscript is now suitable for publication. All changes in the manuscript are marked in red font.

Response to Reviewers

Reviewer 1

1. The introduction is well written and provide the readers with a very comprehensive overview of the issue at hand, also adequately conveying the message about the importance of the present investigation for the understanding of the neurobiological underpinnings of intelligence in humans. I also think the method applied in the present investigation really represents the right tool to answer the Neural Efficiency hypothesis. Congratulations on the insight behind the study design.

Response: We thank the reviewer for the kind words.

2. As previously mentioned, they study has the potential to address a crucial issue with current theories about the neurobiological underpinnings of fluid intelligence in humans. Therefore, it is extremely important that the results do not suffer from any potential bias, especially when it comes to the definition and measurement of the cognitive construct/function of interest. Given the potential impact of the present study, a biased observation might lead to a sequelae of replication studies, as well as others trying to expand such evidence in other directions. This might lead to misleading conclusions and attempts to replicate the present results, which in my opinion are not strictly related to “fluid intelligence”. Specifically, the authors used the BOMAT task, a recently developed matrix-reasoning task thought of measuring gf in humans. Unfortunately, compared to the gold-standard approach for gf assessment, i.e. the matrix reasoning task by Raven (Raven J., Raven J.C., & Court J.H., 1998), the BOMAT suffers from several issues which had brought many researchers to label it more as a working memory task than a pure fluid intelligence one. For instance, Moody clearly address this issue (Moody, 2009) by highlighting how the test, which includes a different parameter space for alternative solutions (i.e. the BOMAT items have a 5*3 grid of alternative solutions = 15) compared to any other matrix-like task,

including the Raven Progressive Matrix (RPM, 3*3) or the Sandia logical/relational test (3*3) (Matzen et al., 2010). Given the increased number of possible answers available to the subject, the BOMAT stresses visuo-spatial working memory significantly more than RPM or Sandia, by reducing the pure contribution of logical reasoning in favor of the ability of holding and manipulating items in each subject's working memory storage. In my personal experience, and please consider it as just an additional evidence, the BOMAT does not measure exactly the same construct as RPM and other matrix reasoning tasks: we have recently analyzed data on ~200 participants who underwent both RPM, Sandia and BOMAT gf assessment, as well as structural and functional MRI acquisitions before and after an executive function training aimed at increasing gf. Factorial analysis has shown how the BOMAT load on a different component with respect to Sandia and RPM, and not surprisingly it shows significant gains after training while neither RPM nor Sandia do. Unfortunately, the selection of gf measures represents a crucial element to push forward a very relevant argument as human intelligence being more related to one or another brain property. If the study gets published and other groups will not be able to replicate the finding by using, for instance, the RPM, it will be difficult to re-label the present findings. To have a first evidence with a more validated measure, or possibly a latent factor level gf score obtained by using multiple gf measures, would be ideal. Specifically, why did the author decide to use the BOMAT instead of the Raven's matrix or other more similar tasks? Do they have other gf tasks that could be used to validate current results? I assume they do not, but if I am wrong I would suggest running the same analysis with the RPM.

Response: Again, we would like to thank the reviewer for the kind words acknowledging the potential impact of our work. We also appreciate the reviewer's in-depth analysis of our methodological approach and agree that the unbiased measurement of the cognitive construct of interest, in this case fluid intelligence, is of utmost importance. To begin with, we would like to discuss the article by Moody (2009) that comments on a study by Jaeggi et al. (2008), in which the BOMAT was used to measure increases in fluid intelligence after participants received training on a working memory task. Moody argues that the observed increases in test performance are hardly surprising due to the fact that the BOMAT consists of a 5*3 matrix configuration, which might draw on visuo-spatial working memory. But Moody also notes, and this is of vital importance, that the BOMAT's emphasis on working memory only came into effect because Jaeggi et al. did not administer the test in the way it is intended to be. They shortened the BOMAT's normally allowed testing time of 45 minutes to merely 10 minutes and thereby "transformed it from a test of fluid intelligence into a speed test of ability to solve the easier visual analogies". We are convinced that the BOMAT is a well-designed test of fluid intelligence, which, due to its larger matrix configuration, might indeed put slightly more emphasis on visuo-spatial working memory compared to other matrix-reasoning tests like Raven's Progressive Matrices. But we also believe that this aspect is negligible if the BOMAT is administered properly without heavy time constraints. Sure enough, we would not go so far as to label the BOMAT a working memory instead of a fluid intelligence test. Our point of view is supported by unpublished data from a study in which both Raven's Advanced Progressive Matrices (RAPM) and the BOMAT were conducted in a sample of 221 university students (not the same sample used for the manuscript under revision). Importantly, the BOMAT was administered with its designated testing time of 45 minutes and not altered in any way. The data clearly show that there is a large and highly significant correlation of $r = .60$ between both test scores, which indicates that the RAPM and the BOMAT measure the same construct, i.e. fluid intelligence. In previous research on

the correlation between the RAPM and another renowned intelligence test, namely the Wechsler Adult Intelligence Scale, correlation coefficients of the same magnitude were reported ($r = .55$; McLaurin et al., 1973).

Furthermore, these data also reveal that the RAPM is likely to suffer from ceiling effects when being conducted in a group of university students. However, the BOMAT was deliberately designed to capture interindividual differences within the ranges of above average intelligence. Thus, the instrument can be used to properly measure fluid intelligence in high potentials and individuals with an academic background. The two histograms below include aforementioned data from 221 university students. As can be seen, the distribution of RAPM test scores is slightly shifted to the right end, whereas the BOMAT test scores are normally distributed.

Due to these observations, we decided to utilize the BOMAT, which measures the same construct as the RAPM, namely fluid intelligence, but is in some parts superior with regard to differentiating in high-IQ ranges. Nevertheless, we completely understand the concerns expressed by the reviewer. Our work features a novel MRI technique which has not been used before to study interindividual differences in fluid intelligence. Since we are breaking new ground here, we are eager to make sure that our findings are as reliable as possible so

that future studies can draw on rock-solid evidence. Unfortunately, we are not able to provide a latent factor of gf since we did not use any tasks other than the BOMAT to assess fluid intelligence. We also believe that it is not feasible to reinvoke all of our 221 participants in order to conduct the RAPM since this would inevitably result in a very high dropout rate. Therefore, we decided to validate our results in another way, namely by using data from the Human Connectome Project (HCP). In so doing, we were not only able to address the concerns expressed by reviewer 1, mainly regarding our approach of fluid intelligence assessment, but could also tackle several issues raised by reviewer 3, which were aimed at our imaging methods. The HCP provides state-of-the-art imaging data and features a multi-shell protocol for the acquisition of diffusion-weighted images. Due to this circumstance, HCP imaging data are compatible with the NODDI technique and allow for the same analyses performed on our original data. In addition to that, the HCP also offers a broad set of behavioral data with various measures of cognitive performance including fluid intelligence. For the purpose of fluid intelligence assessment, the HCP uses the Penn Matrix Analysis Test (PMAT24), a matrix-reasoning test from the Computerized Neuropsychological Test Battery provided by the University of Pennsylvania. This test is based on Raven's Progressive Matrices, which represent the gold-standard approach for gf assessment. The items presented during the PMAT24 have a grid size of 3×3 and thus feature a smaller parameter space for alternative solutions compared to the BOMAT. By validating our original findings with data from another large sample, in which a gf task was conducted that inarguably measures fluid intelligence, we hope to address the concerns expressed by reviewers 1 and 3 and increase the quality of our study in general. We decided to work with the HCP's "S500 plus MEG2" release, which includes about 500 participants, more than twice as many as our original sample. Both samples are comparable with regard to age and sex of the participants. Downloading, preprocessing and analyzing the high-quality data provided by the HCP took an enormous amount of time and effort but paid off in the end. The results obtained by these measures are in line with the ones found in our original data. In order to provide a general overview, we would like to draw the reviewer's attention to Figure 2 and Figure S2 as well as Figure 3 and Figure S4. In both data sets fluid intelligence is positively associated with cortical volume and negatively associated with NODDI coefficients from the overall cortex. Thus, the structure-function relationships found in the HCP data replicate our original results. The combined evidence from both data sets supports our general message that fluid intelligence is enhanced in larger brains with an efficiently organized dendritic microstructure.

3. Overall, the manuscript suggests a strong association between Noddi derived brain properties and fluid intelligence, however the correlation coefficients reported in the manuscript are relatively small ($r = -.19/21$, which mean that only $\sim 4\%$ of variance in intelligence is explained by the different brain features included in the analysis). This is also relevant when the argument about BOMAT is considered: given the very little variance explained by Noddi-indexes, and the potential "gf measurement error", the overall correlation pattern might be actually reflecting noise in the data as well as a relationship between brain properties and an unspecific cognitive construct more related to speed of processing (which constitute a relevant component of BOMAT problem solving, given the higher number of alternative solutions to be tested simultaneously under time pressure) or, as previously mentioned, working memory capacity/processing.

Response: The reviewer is right in that the effect sizes observed in our data are relatively small, and it was not our intention to give a false impression of their magnitude. However, when investigating the association between specific biological markers and complex psychological constructs, one has to expect fairly small correlation coefficients since structure is likely to shape function in a multicausal way. For example, the relationship between brain volume and fluid intelligence has been confirmed multiple times in the past, and was repeatedly quantified by correlation coefficients not larger than .30, comparable to the magnitude of our effects. Given the fact that these associations are fairly small by nature, we agree with the reviewer's opinion that one has to be especially careful to avoid any noise being added to the observations by test instruments of low reliability or validity. In our response to the reviewer's first remark, we already expressed our conviction that the BOMAT, without doubt, measures fluid intelligence instead of working memory capacity or any other related construct. In addition to that, we would like to emphasize that the BOMAT has excellent reliability measures of about .90 as mentioned in the Methods section. Notwithstanding the above, we hope that we were able to ultimately clear up the reviewer's doubts by replicating our results with data from the HCP, for which fluid intelligence was assessed with a different and well-established gf task closer to Raven's Progressive Matrices.

4. Are the correlation results corrected for multiple comparisons?

Response: Following a request made by this reviewer as well as reviewer 4, all correlations were discarded in favor of partial correlations controlling for the effects of age and sex. In addition to that, the partial correlation results that describe associations on the level of single brain regions were FDR corrected for multiple comparisons using the Benjamini-Hochberg method. We revised our description of this approach in the Methods section and made sure that the FDR correction is mentioned properly in all of the respective figures (Figure 4 and Figure S7).

5. In the Result section, the author mention a cortical parcellation they used to look at regional correlation between gf and Noddi measures. Neither information about the nature of the parcellation (structurally or functionally defined regions?) nor the total number of regions (authors mention "34 regions" while the atlas includes 34 regions per hemisphere = 68, see Figure 1) are provided. I would suggest including these information, as well as a rationale for selecting this atlas over many other available ones. Also, I would include such details in the Methods section.

Response: We would like to thank the reviewer for pointing out the slightly confusing description of the utilized cortical parcellation method, which is based on the Desikan-Killiany atlas. We initially decided to use the Desikan-Killiany atlas since its implementation in the FreeSurfer toolbox offers a parcellation of both cortical gray matter and white matter. In line with the explorative nature of our study, this enabled us to investigate the association between fluid intelligence and dendrite microstructure as well as fluid intelligence and axon microstructure on the level of single brain regions. However, since none of the NODDI properties that were averaged across the overall white matter proved to be significantly correlated with fluid intelligence, the analysis of single white matter regions can actually be considered obsolete. Thus, it transpired that a more refined parcellation scheme, even if restricted to the cortex, is the better choice for our work. As a consequence thereof and in reaction to another request by reviewer 3, we decided to replace the Desikan-Killiany

parcellation with the novel multi-modal parcellation provided by the Human Connectome Project (HCPMMP). This parcellation scheme delineates 180 cortical brain regions per hemisphere and is based on the cortical architecture, function, connectivity and topography from 210 healthy individuals. We believe that this considerably more refined parcellation of the cortex is the ideal choice for reporting our results on the level of single brain regions. This becomes particularly evident when comparing regions of interest to the Brodmann areas that constitute the network proposed by the Parieto-Frontal Integration Theory of intelligence (P-FIT). We would also like to clear up any confusion about the number of brain regions being analyzed in the original version of our manuscript. The Desikan-Killiany atlas includes 34 gyral based brain regions per hemisphere. Therefore, FreeSurfer's automated brain segmentation resulted in 68 regions for the overall cortex as well as 68 adjacent regions within the white matter. Since analyzing potential asymmetries between the left and right hemispheres is not within the scope of our work, we decided to average all NODDI markers and volume measures across both hemispheres. Because of this, the 34 brain regions included in our analyses represent mean values derived from two homologous brain regions in the left and right hemispheres respectively. For the revised version of our manuscript, we made sure to describe this statistical approach and the HCPMMP in more detail, hopefully avoiding any confusion.

6. Results section: "Intelligence was significantly associated with ODIcortex in 9 out of 34 brain regions ($r = -.17$ to $-.23$) with most of the identified regions being part of the Parieto-Frontal Integration Theory (P-FIT) of intelligence 7 (Fig. 4A)". Are the regions included in the analysis 34 or 68 (see previous comment)? If the latter (as suggested in Figure 1), are the significant regions 9 per hemisphere or 9 in total? Moreover (see below)

Response: As mentioned above, the 68 regions provided by the Desikan-Killiany atlas were averaged across both hemispheres. This resulted in 34 regions, which we used for all analyses on the level of single brain regions. Likewise, each NODDI coefficient from the 9 statistically significant brain regions represented an average of the values from the left and right hemispheres. However, in the revised version of our manuscript, the in-depth analysis of these 9 significant brain regions was discarded due to the new results that emerged from the use of partial correlation analyses as well as the implementation of the HCP data and its cortical parcellation scheme. The respective results as well as the correspondence between statistically relevant brain regions and the P-FIT model are discussed in the revised version of our manuscript.

7. the authors also suggest that "...most of the identified regions being part of the Parieto-Frontal Integration Theory (P-FIT) of intelligence". How has this been quantified? The atlas used by the authors is the Desikan-Killiany, which has not been used in the original definition of the P-FIT; given the difference among available cortical and subcortical parcellations, it could be that a region labelled as Superior parietal lobe in the present work does not exactly correspond to the same region in the original P-FIT model. This is particularly true for the anatomical parcellation of regions in the prefrontal and parietal lobes where, for instance, atlas like the AAL, Harvard-Oxford and FSL offer completely different parcellation schemes. Did the author computed a Dice coefficient with a P-FIT template?

Response: Taking the correspondence between different parcellation schemes for granted is indeed a potential source of error that one should definitely be aware of. For the original

version of our manuscript, the reported overlap between the statistically significant brain regions from our analyses and the brain regions from the P-FIT model was not quantified in any special way but merely determined by visual inspection. Since this step was supervised by the expertise of Rex Jung, one of the investigators who originally proposed the P-FIT model, we felt that our judgement was sound enough. However, we certainly agree with the reviewer's comment that nothing is more convincing than an argument that can be backed by concrete numbers. Unfortunately, we are not aware of an official P-FIT template that can be used for the sake of pattern matching. Furthermore, the P-FIT model is based on Brodmann areas, which, for the most part, do not correspond to the regions defined by the Desikan-Killiany atlas very well. Thus, a pattern consisting of Desikan-Killiany regions, even when resembling the P-FIT model in the most perfect way possible, would not exhibit a good fit in terms of a Dice similarity coefficient. With regard to these issues, we settled on another method for comparing our result patterns to the P-FIT model. Most importantly, we replaced the Desikan-Killiany atlas with the more refined HCPMMP to perform our analyses on the level of single brain regions. This parcellation scheme is superior to the Desikan-Killiany atlas in that it includes 180 regions per hemisphere instead of just 34, thereby allowing for a better match with the Brodmann areas. The HCPMMP is available in the form of annotation files for FreeSurfer. Using FreeSurfer's `aparc2aseg` tool, we converted these annotation files to a volumetric segmentation of the cortex matching the `fsaverage` standard brain. The same was done to the "`lh.PALS_B12_Brodmann.annot`" and "`rh.PALS_B12_Brodmann.annot`" files included in FreeSurfer, which represent a cortical parcellation based on Brodmann areas created by the Van Essen Lab. As a consequence, each voxel in the HCPMMP segmentation corresponded to a voxel in the Brodmann segmentation. Each region from the HCPMMP segmentation was assigned to a particular Brodmann area depending on the Brodmann area with the highest number of matching voxels. For example, the HCPMMP's V4 region had 782 voxels corresponding to Brodmann area 18 and 2029 voxels corresponding to Brodmann area 19. Thus, the V4 region was finally assigned to Brodmann area 19. By doing so, we were able to determine if a particular region of interest fell into one of the nineteen Brodmann areas constituting the original P-FIT model defined by Jung and Haier or its updated version proposed by Basten et al. (2015). We utilized this method to compare the result patterns which emerged from our own data and the HCP data to the P-FIT model. In both cases, about 70 % of the tested brain regions corresponded to one of the relevant Brodmann areas. The results yielded by this approach as well as a detailed description of the method are included in the revised version of our manuscript.

8. Discussion section: "These results indicate that all of the three brain properties linked to the cortex contribute to the prediction of intelligence independently of each other. In contrast to the results yielded by the correlation analysis, intelligence was no longer significantly associated with ODIWM. Further, we observed no significant associations between intelligence and the remaining predictors INVFWM, VOLWMM, age and sex.". It seems the authors attribute the difference in results to the application of correlation vs regression analyses. Did they try running partial correlations? Results might be similar to those obtained using regression analysis, offering the reader a much more linear sequence of analyses to follow.

Response: We thank the reviewer for this helpful suggestion. As mentioned above, a similar request was made by reviewer 4 and we were happy to implement their ideas in our revised manuscript. All correlations, obtained by analyses on the level of the whole brain or single

brain regions, were discarded in favor of partial correlations controlling for the effects of age and sex. The same approach was used for the respective analyses performed on the HCP data. All results and figures were adapted accordingly with scatterplots now representing the relationships between the residuals of all respective variables.

9.1. Some comments on Autism, brain properties and intelligence in the Discussion section could be toned down. They are relevant to the topic and findings, but given the lack of an equivalent analysis in this patient population using Noddi, I would suggest to limit this reference to a sentence instead of a whole paragraph.

9.2. The same applies to the sentence "This hypothesis is also backed by a PET study in a sample of low-IQ individuals, including patients suffering from Down's syndrome and another form of mental retardation 50. Both low-IQ groups exhibited higher rates of cortical glucose metabolism compared to healthy control participants while working on Raven's Advanced Progressive Matrices. This observation was attributed to a failure of neural pruning in the brains of low-IQ individuals. It is important to note that we worked with mentally healthy subjects and not with a pathological sample. However, there is evidence that the same mechanisms discussed above could also apply to healthy individual with individually different IQs, albeit on a smaller scale." This is not true and the present data does not support this kind of statement. The authors should not compare a significant but very weak evidence (correlation ~ 0.2) in healthy participants with clinical conditions characterized by pathological genetic and developmental substrates. Given this represents the first evidence of a neurite density correlation with intelligence, I would exclude any reference to pathological conditions in the manuscript.

Response: We decided to reference findings obtained from pathological samples since they represent the only data on histologically examined microstructure in a low-IQ population, namely individuals with Down's syndrome and autism. In addition to that, Haier recruited patients suffering from Down's syndrome to serve as a low-IQ group in one of his PET studies on the neural efficiency hypothesis. Consequentially, individuals with Down's syndrome are the only population providing the three types of information that are needed to construct a biological framework for the neural efficiency hypothesis of intelligence. Patients with Down's syndrome are known to have brains with an excess of synapses, they are characterized by their below average IQ and they also tend to show higher rates of cortical glucose metabolism when performing a cognitively demanding task. It was this trifecta of evidence that first enabled Haier to express his idea of higher than normal synaptic densities being the missing link between low intelligence and neural inefficiency. Our own study was inspired by the original work of Haier and we are convinced that the findings he obtained from a pathological sample are to be extended by neuroimaging research on healthy participants. Nevertheless, we agree with the reviewer in that one should always proceed with utmost care when transferring evidence obtained from pathological samples to mentally healthy subjects. A step like this should only be taken when no other source of information are available. Thus, we decided to partly comply with the reviewer's request. We excluded all references to autism from the revised version of our manuscript and tried to tone down the respective paragraphs in which evidence obtained from patients with Down's syndrome is used to support our own findings.

10. Figure 5 is highly speculative. Even though this reflect its main purpose (that is providing the reader with a simplified interpretation of the findings) the nature of the publication should reinforce a more technical depiction of the results. Most importantly, the figure itself suggest a striking –visible to the naked eye– difference between high and low IQ subjects in terms of brain volume, neuron density and arborization of dendritic trees. Instead, the magnitude of the correlations found in the study (e.g. 0.19) is very low (probably corresponding to an effect size of 0.1 or less). I would encourage the authors to carefully edit their manuscript, toning down the overall message and aiming at a more technical report.

Response: The reviewer is correct in that Figure 5 is highly simplified and merely serves as a schematic depiction of our findings. For that purpose, the differences in microstructural properties between high- and low-IQ individuals had to be accentuated. Given the broad readership of a journal like Nature Communications, we would like to stick to this easy-to-understand presentation and are reluctant to apply any major changes to Figure 5. We would like the figure to convey the general message of our study at first glance, namely that less can be more when it comes to the dendritic architecture of intelligent brains. However, the reviewer is completely right that this interpretation has to be put into perspective regarding the magnitude of our effect sizes. Therefore, we highlighted the exaggerated nature of Figure 5 in the figure caption.

Reviewer 2

1. This study aimed to address the fundamental question of what are the biological substrates of intelligence. Previous research has shown that on one hand gray matter volume correlates positively with intelligence, on the other intelligent individuals show reduced brain activity during reasoning. But little is known about the brain tissue microstructure that underlies these observations. This study collected the data that may fill this gap. The central findings of this study, if correct, will clearly represent a major step forward in our understanding of the biological origins of intelligence.

Response: We would like to thank the reviewer for the kind words and are very pleased that the reviewer considers our work a major step forward in understanding the biological origins of intelligence.

2. These findings were drawn based on the data acquired and analysed with a new MRI technique called NODDI. The technique is known to provide three independent markers of tissue microstructure: INVF, which is the neurite density, ODI, which is the neurite orientation dispersion, and ISO, which is the fraction of free water. Interestingly, the authors have chosen to analyze an additional parameter, namely, ENVF, which depends COMPLETELY on INVF, specifically, $ENVF = 1 - INVF$ by definition. They have chosen to analyze this parameter because they argue that it captures the neuron density. This interpretation is a significant departure from what was put forward by the developer of the NODDI technique, which has some unintended consequences: it implies that higher number of neurons (i.e. higher ENVF) can ONLY mean lower number of neurites (i.e. lower INVF), which is probably not warranted. With this in mind, the conclusion concerning the neuron density is questionable. Nevertheless, the conclusion regarding the neurite density and orientation dispersion provides sufficient novelty.

Response: Given our rather unique approach towards the NODDI technique and the unusual implementation of its parameters, we can understand the critical point of view taken by the reviewer. It is entirely correct that the NODDI and AMICO toolboxes only provide three outputs of interest, namely two brain maps representing microstructural environment (ISO, INVF) and one representing microstructural organization (ODI). However, in the original paper by Zhang et al. (2012), the NODDI framework is introduced as a tissue model that distinguishes not two but three types of microstructural environment: CSF, intra-cellular and extra-cellular compartments. The paper also provides a formula that describes how these different environments are represented in the normalized MR signal A.

$$A = (1 - \nu_{iso})(\nu_{ic}A_{ic} + (1 - \nu_{ic})A_{ec}) + \nu_{iso}A_{iso}$$

Here, one can see that the NODDI model calls for a two-step approach. Firstly, the amount of CSF is estimated for a single voxel based on its normalized signal A_{iso} and its volume fraction ν_{iso} . Secondly, the remaining fraction of the voxel, indicated by the $1 - \nu_{iso}$ term, is either assigned to the intra-cellular compartment (INVf) based on its normalized signal A_{ic} and its volume fraction ν_{ic} or the extra-cellular compartment (ENVf) based on its normalized signal A_{ec} and its volume fraction which is written as $1 - \nu_{ic}$. From this it follows that the interdependency of the intra-cellular and extra-cellular environments is embedded in the NODDI framework itself. Consequentially, we believe that computing INVf and ENVf as two parameters being completely dependent on each other is in line with the assumptions made by the developers of the NODDI technique. However, we agree with the reviewer's opinion that it is highly debatable how to interpret a microstructural measure such as ENVf. In the original paper the extra-cellular compartment is defined as "the space around the neurites, which is occupied by various types of glial cells and, additionally in gray matter, cell bodies (somas)". Thus, the magnitude of ENVf is not only shaped by the presence of neurons but also influenced by astrocytes, oligodendrocytes and microglia. Furthermore, capillaries also add to the magnitude of ENVf since they are likely to exhibit hindered diffusion due to being highly permeable. Given the variety of elements constituting the extra-neurite environment, we acknowledge that our original interpretation of ENVf, namely being a direct measure of neuron density, is subject to uncertainty. Additionally, we agree that a scenario in which a higher number of neurites can only mean a lower number of neurons and vice versa is indeed very unlikely. Hence, we decided to follow the reviewer's advice and excluded ENVf as a marker of neuron density from the revised version of our manuscript. As a consequence thereof, we had to adapt certain paragraphs within the Results, Discussion and Methods sections. We also reworked Figure 5 in order to equalize the neuron densities depicted for low- and high-IQ individuals. Despite neuron density being removed as a potential predictor of intelligence, we accord with the reviewer's closing remark that sufficient novelty is provided by the remaining findings regarding neurite density and orientation dispersion.

3. One minor critique concerns the inaccurate description of NODDI as being validated histologically. While there may well be new histological validation of NODDI emerging, the two studies cited (16 and 17) are the validation of an ex vivo MRI technique that the NODDI technique is based on; they are not the validation of NODDI themselves.

Response: We thank the reviewer for pointing this out to us. The reviewer is correct in that only the precursor of the NODDI technique was validated using histological data. At this moment in time, we are not aware of any histological validation of the actual NODDI

technique that is used in our work. Thus, the paragraph in which the two aforementioned studies are cited was rephrased according to the remarks made by the reviewer.

Reviewer 3

1. The authors present interesting results of an inverse correlation between cortical neurite density and intelligence. In general, they use data of moderate quality and use methods that are similarly of moderate quality. There are several additional analyses and methodological improvements that should be easy for the authors to carry out, but should substantially increase the impact of the paper and ensure that confounds are not driving the results.

Response: We are very pleased that the reviewer appreciates the interesting results yielded by our study. We would also like to thank the reviewer for pointing out the numerous methodological improvements that helped to further increase the quality and impact of our study.

2. The parcellation used in this study is the 34-region gyral and sulcal geographic parcellation of FreeSurfer. This parcellation has little to do with brain structure or function, as it is simply based on cortical folding patterns that are known to vary relative to cortical areas. The paper would have higher impact if it used the recently published multi-modal cortical parcellation from the Human Connectome Project. Methods have been released to map this parcellation to fsaverage, and from there the authors could map it to their individual subject surfaces and then to individual subject volumes and on to subject native diffusion space. The authors could then see if any of the cortical areas had particular relationships between NODDI metrics and intelligence. The use of a non-functionally relevant parcellation may be why few of the parcellated statistical tests proved significant.

Response: This has been a very helpful suggestion that led to a substantial improvement of our work. As we already explained in response to a remark made by reviewer 1, we initially decided to use the Desikan-Killiany atlas since its implementation in the FreeSurfer toolbox allows for a parcellation of the cerebral cortex as well as its underlying white matter. Given the fact that we did not observe any significant structure-function relationships for the overall white, we do not see any benefit in reporting an analysis of single white matter regions anymore. Thus, we believe that using a more refined parcellation scheme, even if restricted to the cortex, is indeed a far better choice for our analyses on the level of single brain regions. Since the reviewer also encouraged us to replicate our findings with data from the Human Connectome Project (HCP), we chose to implement its multi-modal parcellation (HCPMMP) as well. Following the steps proposed by the reviewer, we were able to rerun our analyses on the level of single brain regions. This approach provided us with a considerably more detailed pattern of results, which we included in the revised version of our manuscript (Figure 4 and Figure S8).

3. The diffusion data are of moderate spatial resolution (2x2x2mm), this is certainly standard in the field and less than the mean cortical thickness of ~2.6mm. That said, I wondered about the effects of partial voluming when measuring cortical properties. I also wondered about whether cortical thickness and surface area are also correlated with intelligence. Because the authors have run FreeSurfer on all their subjects, it should be trivial to add these parameters to their analyses. In particular I wonder if the multi-variate analysis would

still show effects of these diffusion parameters after cortical thickness differences are accounted for. Also I wonder if these effects would be replicated using the much larger and higher resolution HCP (1.25mm isotropic) diffusion data where behavioral measures of intelligence are also available. I think all of these analyses would increase the impact of this study.

Response: We would like to thank the reviewer for encouraging us to validate our findings with data from the HCP since this measure substantially increased the quality of our study. At first we want to address the reviewer's concerns regarding partial volume effects. The assumption that our effects might be affected by cortical thickness or surface area is valid and has to be investigated further. To this end, we followed the reviewer's recommendation and revisited our analyses by including the respective FreeSurfer parameters. Unsurprisingly, cortical thickness ($r = .18$, $p < .01$) and surface area ($r = .16$, $p < .05$) showed significant correlations with fluid intelligence. This was to be expected since cortical volume was related to thickness ($r = .31$, $p < .001$) and surface area ($r = .93$, $p < .001$) as well. When altering the original multiple regression analysis by replacing cortical volume with thickness, surface area or both parameters, no changes could be observed with regard to the significant influence of the two NODDI markers $INVF_{Cortex}$ and ODI_{Cortex} . Furthermore, neither cortical thickness nor surface area showed statistically significant contributions of their own in predicting fluid intelligence.

	β	p
$INVF_{Cortex}$	-0.24	0.02*
$INVF_{WM}$	0.13	0.16
ODI_{Cortex}	-0.26	0.00**
ODI_{WM}	0.08	0.38
THICK	0.11	0.13
VOL_{WM}	0.14	0.10
SEX	-0.06	0.45
AGE	-0.05	0.48

	β	p
$INVF_{Cortex}$	-0.25	0.00**
$INVF_{WM}$	0.15	0.12
ODI_{Cortex}	-0.25	0.01**
ODI_{WM}	0.07	0.46
SURF	0.15	0.29
VOL_{WM}	-0.01	0.92
SEX	-0.06	0.44
AGE	-0.06	0.47

	β	p
$INVF_{Cortex}$	-0.23	0.02*
$INVF_{WM}$	0.13	0.16
ODI_{Cortex}	-0.25	0.01**
ODI_{WM}	0.09	0.35
THICK	0.11	0.14
SURF	0.15	0.31
VOL_{WM}	0.02	0.91
SEX	0.62	0.61
AGE	0.08	0.60

Multiple regression analyses predicting fluid intelligence as measured by the BOMAT. $INVF_{Cortex}$, $INVF_{WM}$, ODI_{Cortex} , ODI_{WM} , VOL_{WM} SEX and AGE represent the same independent variables used in the original analysis from our manuscript. Importantly, VOL_{Cortex} was either replaced by cortical thickness (THICK, left table), cortical surface area (SURF, middle table) or both parameters at once (right table). The last two columns of each table give the beta-weights and p-values for each predictor. * $p < 0.05$, ** $p < 0.01$

Since cortical thickness and surface area apparently do not affect our findings in any substantial way, we decided not to mention the two parameters in the revised version of our manuscript for the sake of clarity. Instead, we retained cortical volume as our morphometric measure of choice as it is very easy to interpret. In addition to that, cortical volume is likely to be the most common parameter used to quantify brain size in previous research. With regard to the reviewer's second remark, concerning the replication of our findings using data from the HCP, we would like to draw the reviewer's attention to our first response to reviewer 1. In the last paragraph we give a detailed description of our approach to implementing the HCP data into our work.

4. I also wondered about the effect of motion on these measures. Might more intelligent subjects also be more cooperative in the scanner, moving less relative to less intelligent subjects who might move more and have biases in their structural or diffusion data? I think a summary measure of motion (e.g. number of volumes above a frame-wise displacement (FD) threshold of X or one of the many other summary motion measures) should be included in the analyses as a covariate of no interest.

Response: The assumption that fluid intelligence is related to compliance and head motion in the scanner sounds valid. Given its potential impact on our results, we decided to perform the suggested analyses. To this end, we used a summary measure of head motion that was calculated as the average of the relative displacement values yielded by the eddy_correct tool implemented in FSL. In our data, fluid intelligence is not significantly correlated with head motion measured during the diffusion-weighted scan ($r = -.02$, $p = .80$). Consequentially, adding head motion as a covariate to the partial correlations or the multiple regression analysis does not alter the respective results in any particular way. When predicting fluid intelligence in combination with age, sex and all brain properties of interest, the beta-weight of head motion turns out to be negligibly small and is far from being statistically significant ($\beta = .04$, $p = .57$). In addition to that, the significant influence of VOL_{Cortex} and the two NODDI markers $INVF_{Cortex}$ and ODI_{Cortex} remains untouched.

	β	p
INVF_{Cortex}	-0.24	0.02*
INVF_{WM}	0.15	0.12
ODI_{Cortex}	-0.24	0.01**
ODI_{WM}	0.08	0.37
VOL_{Cortex}	0.23	0.05*
VOL_{WM}	-0.04	0.73
SEX	-0.03	0.77
AGE	-0.04	0.62
MOTION	0.04	0.57

Multiple regression analyses predicting fluid intelligence as measured by the BOMAT. $INVF_{Cortex}$, $INVF_{WM}$, ODI_{Cortex} , ODI_{WM} , VOL_{Cortex} , VOL_{WM} SEX and AGE represent the same independent variables used in the original analysis from our manuscript. Importantly, head motion (MOTION) was added as another independent variable. The last two columns of each table give the beta-weights and p-values for each predictor. * $p < 0.05$, ** $p < 0.01$

Since we found the relationship between fluid intelligence and head motion to be of no importance to the analysis of our data, we decided to not mention it in the manuscript. Given the many other variables included in the study, we believe that this approach enhances the readability of the manuscript and facilitates understanding of the results.

5. The authors don't really explain why ODI is negatively correlated with intelligence. One explanation is that ODI in the cortex is driven largely by partial volume effects (or motion might be another good cause) and thinner cortex would have more partial volume effects (and might also be associated with lower intelligence). This gets back to point B where it would be helpful to see if the ODI effect remains after partialling out cortical thickness variability.

Response: If we understand correctly, the reviewer suggests an interesting chain of causality in which low intelligence is linked to increased head motion, more pronounced image distortions, thinner cortical ribbons, larger partial volume effects and finally increased ODI in the cortex. It is true that head motion in the scanner is known to cause image distortions with cortical ribbons being displayed thinner than they usually are. Given a fixed spatial resolution, this scenario would lead to larger partial volume effects at the gray and white matter boundary. In turn, the average ODI in the cortex would be decreased due to the below average ODI in the white matter. Assuming that head motion occurs in participants with low compliance due to low intelligence, the outcome would be a positive correlation between fluid intelligence and ODI in the cortex. However, our results show a different pattern with fluid intelligence being negatively associated with ODI in the cortex. As stated in our manuscript, we believe that this negative correlation can be explained as follows. It is conceivable that both INVF and ODI are linked to the amount of synaptic connections formed by dendritic trees. According to previous research, an increase in synapses causes failure in differentiating signals from noise and thus reduces network efficiency. In addition to that, computational studies show that synaptic pruning not only increases learning and processing speed but also saves network and energy resources. Thus, we are convinced that low ODI coefficients are more likely to be found in individuals with high intelligence because they represent less arborized dendritic trees with synaptic connections reduced to an efficient minimum. For the purpose of conveying this line of argumentation to the reader more clearly, we decided to slightly adapt the caption of Figure 5, which visualizes the tissue properties quantified by INVF and ODI. Notwithstanding these clarifications above, we hope that having successfully validated our results with data from the HCP is convincing enough for the reviewer to overcome his doubts regarding potential shortcomings of our diffusion-weighted data.

6. It is necessary to estimate the intrinsic free diffusivity parameter for greymatter instead of just using the parameters for white matter and CSF e.g. using the procedure introduced by Guerrero et al. (2016) 'Investigating the effects of intrinsic diffusivity on neurite orientation dispersion and density imaging (NODDI)' (<http://dev.ismrm.org/2016/1046.html>). Was this done in this study?

Response: In order to compute the NODDI coefficients that were analyzed in our study, we adopted the AMICO framework developed by Daducci et al. (2015). As to our knowledge, separately estimating the intrinsic free diffusivity parameter for gray matter is not part of this tool. Given the large samples constituting our own data as well as the HCP data, using the AMICO framework offered the possibility to speed up processing considerably. AMICO is also the method of choice utilized by other labs working with large samples. For example, the renowned FMRIB group at the University of Oxford recently published a paper in Nature Neuroscience covering the UK Biobank project which includes more than 5000 subjects. Their approach towards using the AMICO framework is comparable to ours in that they did not compute separate estimates of the intrinsic free diffusivity parameter for different types of tissue. Thus, we believe that our results can be regarded as equally valid.

7.1. Awkward phrasing: Page 3 line 46 "has been met with" (in general the manuscript would benefit from further editing for typos and style)

7.2. Please give the range and standard deviation of the participants ages.

7.3. Please give the mean, range, and standard deviation of the intelligence scores being correlated with NODDI.

7.4. It is not necessary to call things “highly significant” just because they have p-values that are more than a little bit lower than 0.05. Perhaps such hyperbole should be reserved for those things that have very small p-values requiring scientific notation to display.

7.5. Page 7 line 147: “designed an integrative neural prediction model” this seems like an unnecessarily complicated description of a basic (and appropriate) multi-variate analysis

7.6. Page 7 line 151: typo “included”

7.7. Page 9 line 181: awkward phrasing: “It was the overarching goal of this study” to “The overarching goal of this study was...”

7.8. Page 9 line 198: awkward phrasing: “bring forth stick-like diffusion patterns from which” to “are modeled as sticks in the NODDI model, from which NODDI markers resembling...”

Response: We appreciate the effort with which the reviewer inspected our manuscript in order to point out these little but important mistakes to us. All typos and awkwardly phrased sentences were corrected accordingly. The mean, range and standard deviation of the participants' age and intelligence scores were also added to the revised version of our manuscript.

8. There is no mention in the methods of how the diffusion data were registered to the T1w data, nor of how b0 distortions in the diffusion data were removed so that this registration was accurate. A very accurate registration is necessary so that the cortical ribbon in the diffusion data lines up with the cortical ribbon of the structural data – even being off by a few mm will lead to a major mismatch as the cortex is only 2.6mm thick on average. Also it is not clear if eddy or eddy_correct was used to do motion and eddy current correction. Eddy is the preferred software tool for this.

Response: We share the reviewer's point of view regarding the importance of accurate image registration. It is entirely correct that mismatches between the diffusion-weighted and structural data might affect the outcome in a negative way, especially in structures as thin as the cerebral cortex. As for our data, image registration was performed utilizing the bbregister tool in FreeSurfer. The quality of each registration was carefully checked by plotting the outline of the registered image's white matter against the respective diffusion-weighted and structural images. In order to meet the reviewer's request, we included a more detailed description of this registration procedure in the Methods section of our revised manuscript. Furthermore, we also reported the eddy_correct tool as being our method of choice for motion and eddy current correction. As to our knowledge, the eddy tool, for its advantages to fully come into effect, requires either the output generated by the topup tool or another fieldmap representing the susceptibility induced off-resonance field. Unfortunately, neither was available to us since the volumes of our diffusion-weighted data were only acquired in the anterior-posterior phase-encoding direction. Thus, we were compelled to use the eddy_correct tool to perform motion and eddy current correction.

Since the reviewer stated that eddy is the preferred tool for these processing steps, we were once again encouraged to validate our findings with data from the HCP. For this purpose, we decided to utilize diffusion-weighted data from the HCP's "S500 plus MEG2" release which is of very high quality in respect of both image acquisition and preprocessing. Comparable to our approach, preprocessing of structural data, including surface reconstruction and automated brain parcellation, was carried out with the FreeSurfer pipeline. Diffusion-weighted data was corrected for head motion, eddy currents and EPI distortions using the aforementioned eddy tool in FSL. Here, we also made sure to provide an elaborate description of these preprocessing steps in the Methods section of our revised manuscript.

Reviewer 4

1. There are a number of strengths of this study. The question is interesting in general, the authors apply a multi-shell diffusion scheme on a large sample appropriate for using the NODDI reconstruction. That reconstruction algorithm is itself novel and innovative. Methodologically the study is very sound. The multi-shell acquisition has sufficient directions (120) and shells (3) to do the NODDI reconstruction, and the resolution is good (2mm isotropic). Reconstruction steps were all appropriate and consistent with standards in the field. The use of a normed and reliable (split-half = .89) matrix reasoning test to assess intelligence is reasonable.

Response: We are very thankful that the reviewer acknowledges the numerous strengths of our study and appreciate the kind words.

2. My major concern is whether the authors, in the correlation analysis, are p-hacking their way to significant effects, and those effects are very small. I don't mean this in a derogative way—it is important to explore the data. However, if the effects simply pop out of running a large number of correlations, that is another matter. For example, the main correlations with the whole brain measures have effect sizes of $r = .19$ and $r = .14$. The data from Figure 2 are not very convincing that there is a real effect in terms of the size, although both are significant. But would they replicate? It is not clear these were FDR corrected—the text is ambiguous. It says "Testing was two-tailed with an α -level of .05 which was FDR corrected for multiple comparisons when conducting correlation analyses on the level of single brain regions". This seems to imply that FDR correction was not applied to the whole-brain analysis, although the authors could clarify this, as I may be misreading.

Response: In common with reviewer 1, this reviewer is concerned about the replicability of our findings. As pointed out in one of the answers to a comment made by reviewer 1, small correlation coefficients are to be expected when investigating the associations between specific biological markers and complex psychological constructs. Nevertheless, we agree that one has to be especially careful about these small effect sizes representing real structure-function relationships instead of merely being produced by statistical noise. Therefore, we implemented several measures to avoid reporting false positive results. Firstly, the whole-brain analyses were not corrected for multiple comparisons but with the revised version of our manuscript we decided to discard the original correlations in favor of partial correlations controlling for the effects of age and sex. Furthermore, the pattern of results which emerges from these partial correlations is also to be found in the multiple regression analysis. Here, the unique contributions of cortical INVF, ODI and VOL towards

fluid intelligence remain statistically significant despite being tested together with all of the relevant white matter properties as well as age and sex. Lastly, the correlations on the level of single brain regions were changed to partial correlations as well and, as the reviewer pointed out correctly, FDR correction was applied to all of them. We believe that our findings are already based on a sound statistical fundament due to the aforementioned measures. However, this reviewer's critical comment on our result's replicability as well as remarks made by reviewers 1 and 3, led us to add another layer of safety and rerun our analyses with data from the Human Connectome Project (HCP). In doing so, we were able to show that our findings can be confirmed by an independent data set with an even bigger sample, imaging data of the best available quality and a different test for fluid intelligence. We would like to thank the reviewer for challenging us to put in the additional expenditure. The results, which emerged from successfully replicating our results with the HCP data, were included in the revised version of our manuscript.

3. The regression analysis is more convincing, and my opinion is that the authors should skip reporting the correlation analysis. Correcting for age and sex seems important. The authors could report on co-linearity, but they were careful to leave out ENFV so it is likely they examined this. In addition, the R-squared value should be reported. Is it explaining a lot of variance? The advantage here is we see 1) the slightly larger effect sizes; 2) the fact that age and sex are not significant predictors.

Response: We agree with the reviewer's opinion that analyzing our data by means of multiple regression analysis is most convincing in terms of statistical significance. Nevertheless, for the purpose of providing the reader with a first overview, we find that simple correlations showing the general pattern of structure-function relationships serve as an adequate tool. Thus, we decided to keep the correlations as a first step within our analyses. However, as mentioned above, we reacted to the reviewer's proposal by discarding all correlations in favor of partial correlations correcting for the effects of age and sex. The same was done to the correlation analyses performed on the level of single brain regions. All of the respective R^2 and p values were added to the revised version of our manuscript accordingly.

4. An added variable plot for the whole brain analysis may also be more convincing, and this could replace Figure 2. [...] In combination with Figure 3, this would show visually how well the data show the linear relation, controlling for other variables. If the regression is applied to Figure 4, the same presentation could be made (leaving out age and sex).

Response: We would like to thank the reviewer for suggesting a more appropriate way of visualizing our statistical findings. In reaction to the reviewer's remark, we decided to change all scatterplots representing simple correlation analyses to added variable plots now representing the newly implemented partial correlation analyses controlling for the effects of age and sex (Figures 2, S1, S2, S3). In addition to that, the regression analyses performed on our original data set as well as data from the HCP were also visualized by means of added variable plots and included as Supplementary Information in the revised version of our manuscript (Figure S5 and Figure S6).

5. Finally, to try to avoid fishing for findings, the authors may want to compute the measures for brain regions that would be very unlikely to be associated with intelligence, and show

that they are not. For example, would metrics in primary cortical regions be associated with intelligence? What about cerebellum?

Response: This is an interesting suggestion. However, given the novelty of our imaging technique and its application within our study, namely investigating the association between fluid intelligence and microstructural brain properties, our approach happens to be explorative by nature. Therefore, we felt the need to direct our analyses at the entirety of the cerebral cortex instead of restricting them to particular regions defined by a priori assumptions. As we already pointed out in our answer to the reviewer's first remark, we properly addressed the possibility of false discoveries by correcting our results for multiple comparisons when performing any analyses at the level of single brain regions. Thus, we consider our approach justified and do not think it is being prone to be fishing for findings. Nevertheless, since replicating the results found on the level of single brain regions was of major importance to reviewer 3 as well, we decided to rerun our analyses with data from the HCP and to use its more refined multi-modal parcellation scheme. We are confident that this approach might clear up any of the reviewer's remaining concerns about the lack of a priori assumptions guiding our choice of brain regions to be analyzed. As a side note, the cerebellum was already shown to be associated with fluid intelligence despite being a very unlikely candidate for this. According to Ramsden et al. (2011), gray matter density within the anterior cerebellum is related to non-verbal IQ.

6. The authors repeatedly write that larger brains = more neurons, but this is not necessarily the case because the type/size of neuron matters, and the density (e.g., there are more neurons in the cerebellum than the cortex). Perhaps they could correct this.

Response: The reviewer is correct in that the number of neurons might vary drastically across different parts of the brain, depending on neuron type or size. The cerebellum is a very good example to prove this point since it includes about four times as many neurons compared to the cerebral cortex despite possessing a mere fraction of its mass. However, such differences in neuron composition are negligible unless one wants to compare two tissue samples of the same volume but from different parts of the brain. This is not what we did in this study nor does our statement about larger brains possessing more neurons imply such a scenario. It is true that in some special cases a small brain might possess more neurons than a big brain due to certain parts of the small brain, like the cerebellum, being more accentuated with regard to relative volume. Nevertheless, given the fairly large correlation between neocortical volume and total number of neurons ($r = .71$, Pakkenberg & Gundersen, 1997), one has to acknowledge that such special cases can be neglected and larger brains are very likely to possess more neurons compared to smaller ones.

Reviewers' comments:

Reviewer #2 (Remarks to the Author):

The revised manuscript has addressed the concerns that I have raised.

I do have a few additional, albeit minor, comments that I'd like to draw the authors' attention to.

1. In the revised text, the authors have clarified the processing of their own dataset. In particular, it is now revealed that `eddy_correct`, an out-dated tool, was used to pre-process their own diffusion data. Some authors have recently shown that the tool is not well-suited for pre-processing multi-shell diffusion dataset used here (<http://www.sciencedirect.com/science/article/pii/S1053811915010289>). The replication of the findings with the HCP data, which used the modern version of the tool, suggests that any bias from using `eddy_correct` is not sufficient to significantly confound the results. Nevertheless, for completeness, it might be useful for the authors to acknowledge this limitation in the discussion.
2. Figure 1, which shows the NODDI parameter maps, can benefit from including grayscale bars, given that these parameters have definite values. The intensity windowing can benefit from some adjustment to better reveal contrast in gray matter.
3. Figure 5 needs to have a clear warning that the illustration is a significant over-exaggeration, as the actual differences revealed by the study are far more subtle.
4. While I had pointed out a lack of direct histological validation of NODDI in my first round of comment, I need to acknowledge that my earlier statement is no longer accurate. A histological validation of NODDI was recently published: <http://onlinelibrary.wiley.com/doi/10.1002/acn3.445/full>

Reviewer #3 (Remarks to the Author):

The authors have substantially improved their manuscript by taking a number of my and the other reviewers' suggestions. That said, there are still some outstanding concerns:

- 1) The authors find no relationship with head motion, which is reassuring; however, on the other hand they don't want to mention that they checked for this in the manuscript. I think they should mention the motion analysis, as other readers will likely wonder about motion effects and it would be much better to have a supplemental analysis showing that there is no issue than saying nothing at all about the matter in the paper.
- 2) The authors did not want to separately optimize the NODDI model for grey matter. Unfortunately this is necessary according to the author of the NODDI model because otherwise the ODI values will be biased in grey matter. This is also discussed in an

independent ISMRM 2016 abstract by Guerrero et al, which is available from the editor. We have optimized the intrinsic free diffusivity for grey matter in the HCP data according to the Guerrero et al approach and the value is $1.1 * 10^{-3} \text{ mm}^2 / \text{s}$ (as opposed to the white matter optimized default of $1.7 * 10^{-3} \text{ mm}^2 / \text{s}$). I would recommend rerunning the NODDI model with the grey matter optimized parameter for the greymatter analyses.

3) Many of the replications of the results are impressive, but it is notable that the multiple regression model for ODI does not replicate, although the partial correlation does. Is this because in the HCP data ODI and cortical volume are not uniquely variable? I was also disappointed to see that the areal analyses did not replicate in terms of specific cortical areas that were statistically significant either. Is this because of differences in the MRI data or the cognitive test between the two datasets? The authors used only half of the HCP data, so perhaps the rest would at least match the S499 result?

4) The kinetikor system didn't end up being used for anything but the earliest scans in the HCP study for technical reasons, so I would delete that sentence from the methods.

5) The number of diffusion directions should be stated for the HCP study.

6) It is not necessary to convert surface label files to volume files to compute overlap, however if the same surface is used for surface to volume mapping, it will not matter.

7) Ideally the HCP parcellation's color scheme would be preserved, rather than using the random scheme currently used in Figure 1.

Reviewer #4 (Remarks to the Author):

The authors have adequately dealt with my initial critiques. The addition of a replication data set provides an additional check on the previously reported results, which improves the quality of the manuscript.

Response to Reviewers:
**Diffusion markers of dendritic density and arborization in gray matter
predict differences in intelligence**

Erhan Genç, Christoph Fraenz, Caroline Schlüter, Patrick Friedrich, Rüdiger Hossiep, Manuel
C. Voelke, Josef M. Ling, Onur Güntürkün and Rex E. Jung

General Remarks

Three of the original reviewers were invited to evaluate the revised version of our manuscript and all of them appreciated the changes we made to address their concerns. Likewise, we agree that the quality of our work has benefitted greatly from this first round of revisions. Thus, we would like to thank all four reviewers for their helpful comments. Nevertheless, as pointed out by the reviewers, there were still some important points to be addressed. To summarize, Reviewer 2, while generally satisfied with our revised manuscript, encouraged us to make some minor additions to the text and figures. We gladly implemented all of these suggestions. Reviewer 3 also requested some small changes to the text and figures but was mainly concerned about our analyses assuming equal values of intrinsic diffusivity for gray and white matter microstructure. We followed the reviewer's advice and utilized the parameters suggested by Guerrero et al. to rerun the AMICO pipeline for both data sets now including a total of 765 subjects. By analyzing the data with the NODDI model optimized for gray matter, we were able to confirm the main results from our revised manuscript. Finally, Reviewer 4 was completely satisfied with our first response to the initial critiques and acknowledged the improvement in the manuscript's quality due to the addition of a replication data set. We believe that our second round of revisions, including the entire reanalysis of data with an optimized NODDI model, substantially increased the quality and impact of this work even further. All issues raised by the reviewers were addressed in a careful manner and the respective changes are highlighted in red font within the text file. We hope the manuscript is now suitable for publication.

Response to Reviewers

Reviewer 2

The revised manuscript has addressed the concerns that I have raised. I do have a few additional, albeit minor, comments that I'd like to draw the authors' attention to.

1. In the revised text, the authors have clarified the processing of their own dataset. In particular, it is now revealed that `eddy_correct`, an out-dated tool, was used to pre-process their own diffusion data. Some authors have recently shown that the tool is not well-suited for pre-processing multi-shell diffusion dataset used here (<http://www.sciencedirect.com/science/article/pii/S1053811915010289>). The replication of the findings with the HCP data, which used the modern version of the tool, suggests that any bias from using `eddy_correct` is not sufficient to significantly confound the results. Nevertheless, for completeness, it might be useful for the authors to acknowledge this limitation in the discussion.

Response: We are very pleased to read that our first round of revisions was able to address all major concerns raised by the reviewer. Regarding this still remaining, albeit minor, request, we now mention our use of `eddy_correct` and its limitations in the discussion section and added the respective article to our reference list.

2. Figure 1, which shows the NODDI parameter maps, can benefit from including grayscale bars, given that these parameters have definite values. The intensity windowing can benefit from some adjustment to better reveal contrast in gray matter.

Response: This is a useful suggestion and we gladly added grayscale bars to each of the NODDI parameter maps in Figure 1. We also tried to adjust the contrast of each map in order to emphasize intensity variations within the cortical ribbon. However, given the rather small depiction of cortical gray matter within Figure 1, our efforts did not bring about significant changes visible to the naked eye. Thus, we decided to stick with the original intensity windowing since it is well-suited to highlight the differences in NODDI values between gray and white matter compartments. We believe that this kind of visualization provides the best overview on NODDI parameters and their variation across the brain, especially for readers who are unfamiliar with the method.

3. Figure 5 needs to have a clear warning that the illustration is a significant over-exaggeration, as the actual differences revealed by the study are far more subtle.

Response: This is an important point that was also mentioned by Reviewer 1 in reaction to our first version of the manuscript. In order to address this issue we decided to highlight the exaggerated nature of the figure in its caption.

4. While I had pointed out a lack of direct histological validation of NODDI in my first round of comment, I need to acknowledge that my earlier statement is no longer accurate. A histological validation of NODDI was recently published: <http://onlinelibrary.wiley.com/doi/10.1002/acn3.445/full>

Response: This is a suitable publication to highlight the validity of NODDI. We gladly added it to the reference list of the revised manuscript.

Reviewer 3

The authors have substantially improved their manuscript by taking a number of my and the other reviewers' suggestions. That said, there are still some outstanding concerns:

1. The authors find no relationship with head motion, which is reassuring; however, on the other hand they don't want to mention that they checked for this in the manuscript. I think they should mention the motion analysis, as other readers will likely wonder about motion effects and it would be much better to have a supplemental analysis showing that there is no issue than saying nothing at all about the matter in the paper.

Response: Since we decided to rerun the AMICO pipeline with optimized parameters and obtained comparable but slightly different results, it was necessary to check for confounding effects of head motion once again. As with our previous analysis, head motion was not

associated significantly with any variables of interest. We followed the reviewer's advice to inform the readers about these results and included the supplemental analysis in the revised version of our manuscript.

2. The authors did not want to separately optimize the NODDI model for grey matter. Unfortunately this is necessary according to the author of the NODDI model because otherwise the ODI values will be biased in grey matter. This is also discussed in an independent ISMRM 2016 abstract by Guerrero et al, which is available from the editor. We have optimized the intrinsic free diffusivity for grey matter in the HCP data according to the Guerrero et al approach and the value is $1.1 * 10^{-3} \text{ mm}^2 / \text{s}$ (as opposed to the white matter optimized default of $1.7 * 10^{-3} \text{ mm}^2 / \text{s}$). I would recommend rerunning the NODDI model with the grey matter optimized parameter for the greymatter analyses.

Response: We addressed this point and reran our analyses using the optimized model suggested by the reviewer. The computation of NODDI parameter maps was carried out using the AMICO toolbox with the intrinsic free diffusivity value being fixed to $1.1 * 10^{-3} \text{ mm}^2 / \text{s}$ in gray matter and $1.7 * 10^{-3} \text{ mm}^2 / \text{s}$ in white matter compartments. The pattern of results that we obtained in doing so showed a substantial overlap with our findings from the original analyses. The general outcome of our partial correlation and multiple regression analyses was not affected in any significant way. We merely observed slight changes in the correlations between NODDI parameters derived from the data provided by the Human Connectome Project (see next point). Therefore, our findings once again proved to be solid. We implemented the newly obtained results in the revised version of our manuscript and added a description of our approach using the optimized NODDI model to the methods section.

3. Many of the replications of the results are impressive, but it is notable that the multiple regression model for ODI does not replicate, although the partial correlation does. Is this because in the HCP data ODI and cortical volume are not uniquely variable? I was also disappointed to see that the areal analyses did not replicate in terms of specific cortical areas that were statistically significant either. Is this because of differences in the MRI data or the cognitive test between the two datasets? The authors used only half of the HCP data, so perhaps the rest would at least match the S499 result?

Response: We agree with the reviewer in that it is indeed impressive to see our findings being replicated by the Human Connectome Project's data in such substantial manner, given that both data sets are very different on many levels. For example, the reviewer mentions the two different cognitive tests that were used in order to measure intelligence, i.e. BOMAT and PMAT24. In the first round of revisions we already elaborated on the similarities between these tests. Both of them are culture-fair matrix-reasoning instruments capable of assessing the construct of fluid intelligence. But they also differ in important aspects such as the distribution of test scores within high-IQ ranges. The BOMAT was deliberately designed to avoid ceiling effects, whereas the PMAT24, since being composed of items from Raven's Progressive Matrices, is likely to estimate IQ distributions that are slightly skewed to the left. The reviewer also brought up the obvious differences in MRI data. In the first round of revisions the quality of our data was acknowledged by Reviewer 4 who stated that our "multi-shell acquisition has sufficient directions and shells to do the NODDI reconstruction, and the resolution is good". However, it goes without saying that the data provided by the

Human Connectome Project is of outstanding quality and superior in terms of data acquisition and preprocessing. As a consequence thereof, we see differences in voxel size (2x2x2 mm vs. 1.25x1.25x1.25 mm) and number of total diffusion directions (128 vs. 288) as well as in preprocessing protocols (e.g. eddy_correct vs. eddy). Another important aspect worth mentioning is that the two samples themselves are not completely equal to one another. Our data set includes 259 participants with about 53% being male, whereas the sample we utilized from the "S500 plus MEG2" release features almost twice as much participants (N = 498) of which merely 41% are males. Given all these differences, it is hardly surprising that there are some results from our data that do not exactly match those obtained from the Human Connectome Project. Nevertheless, we feel that the similarities far outweigh the minor differences. Both data sets indicate that intelligence is associated with neurite density and orientation dispersion. Equally important, both data sets also show that this association points into a negative direction. This general pattern is clearly visible in both data sets. Moreover, one has to acknowledge that most of the statistically significant cortical areas, despite lacking a perfect match between data sets, show an impressive overlap with regions previously identified as belonging to the P-FIT network (about 70%). As requested by Reviewer 2, we decided to point out the differences in preprocessing protocols used for the S259 and S498 samples in the revised version of our manuscript. In addition to that, we will also elaborate on the various other differences between the two data sets described above. The reviewer noted that the multiple regression model for ODI does not replicate, although its partial correlation does. We would like to provide some additional data on the reviewer's assumption that ODI and cortex volume might not be uniquely variable in the Human Connectome Project's data.

S221 sample - $1.7 * 10^{-3} \text{ mm}^2/\text{s}$				
	$\text{VOL}_{\text{Cortex}}$	$\text{INVF}_{\text{Cortex}}$	$\text{ODI}_{\text{Cortex}}$	$\text{ISO}_{\text{Cortex}}$
$\text{VOL}_{\text{Cortex}}$	-	-	-	-
$\text{INVF}_{\text{Cortex}}$	-0.11 (.09)	-	-	-
$\text{ODI}_{\text{Cortex}}$	-0.02 (.82)	.10 (.16)	-	-
$\text{ISO}_{\text{Cortex}}$	-0.18 (.01)	.22 (.00)	-0.24 (.00)	-

S259 sample - $1.1 * 10^{-3} \text{ mm}^2/\text{s}$				
	$\text{VOL}_{\text{Cortex}}$	$\text{INVF}_{\text{Cortex}}$	$\text{ODI}_{\text{Cortex}}$	$\text{ISO}_{\text{Cortex}}$
$\text{VOL}_{\text{Cortex}}$	-	-	-	-
$\text{INVF}_{\text{Cortex}}$	-0.12 (.06)	-	-	-
$\text{ODI}_{\text{Cortex}}$	-0.19 (.00)	.11 (.07)	-	-
$\text{ISO}_{\text{Cortex}}$	-0.31 (.00)	.40 (.00)	.01 (.90)	-

Summary of correlation coefficients between cortical volume and NODDI parameters derived from the cortex. Left table shows data from the S221 sample with intrinsic free diffusivity being fixed to $1.7 * 10^{-3} \text{ mm}^2/\text{s}$ (default setting). Right table shows data from the S259 sample with intrinsic free diffusivity being fixed to $1.1 * 10^{-3} \text{ mm}^2/\text{s}$ (optimized model). $\text{ODI}_{\text{Cortex}}$ correlations are marked in light gray.

S499 sample - $1.7 * 10^{-3} \text{ mm}^2/\text{s}$				
	$\text{VOL}_{\text{Cortex}}$	$\text{INVF}_{\text{Cortex}}$	$\text{ODI}_{\text{Cortex}}$	$\text{ISO}_{\text{Cortex}}$
$\text{VOL}_{\text{Cortex}}$	-	-	-	-
$\text{INVF}_{\text{Cortex}}$.02 (.74)	-	-	-
$\text{ODI}_{\text{Cortex}}$	-0.04 (.43)	.45 (.00)	-	-
$\text{ISO}_{\text{Cortex}}$	-0.04 (.33)	.20 (.00)	.13 (.00)	-

S498 sample - $1.1 * 10^{-3} \text{ mm}^2/\text{s}$				
	$\text{VOL}_{\text{Cortex}}$	$\text{INVF}_{\text{Cortex}}$	$\text{ODI}_{\text{Cortex}}$	$\text{ISO}_{\text{Cortex}}$
$\text{VOL}_{\text{Cortex}}$	-	-	-	-
$\text{INVF}_{\text{Cortex}}$	-0.03 (.57)	-	-	-
$\text{ODI}_{\text{Cortex}}$	-0.03 (.50)	.59 (.00)	-	-
$\text{ISO}_{\text{Cortex}}$.03 (.52)	.53 (.00)	.28 (.00)	-

Summary of correlation coefficients between cortical volume and NODDI parameters derived from the cortex. Left table shows data from the S499 sample with intrinsic free diffusivity being fixed to $1.7 * 10^{-3} \text{ mm}^2/\text{s}$ (default setting). Right table shows data from the S498 sample with intrinsic free diffusivity being fixed to $1.1 * 10^{-3} \text{ mm}^2/\text{s}$ (optimized model). $\text{ODI}_{\text{Cortex}}$ correlations are marked in light gray.

The two tables above, featuring data from the old S499 sample as well as the new S498 sample, clearly show that there is no significant correlation between $\text{VOL}_{\text{Cortex}}$ and $\text{ODI}_{\text{Cortex}}$ in the Human Connectome Project's data. However, one can observe some strong associations between the three NODDI parameters in both data sets, especially with the intrinsic free

diffusivity value being fixed to $1.1 * 10^{-3} \text{ mm}^2/\text{s}$. Due to this increased multicollinearity, brought about by the NODDI model optimized for gray matter, it is conceivable that the different NODDI parameters now share a fair amount of explained variance when predicting intelligence. We decided to address this issue in two ways. First, we added Table S1 and Table S2 to the revised version of our manuscript in order to provide the reader with a clear overview on correlations among our variables. Second, we followed a more stringent approach in our analyses on the level of single brain regions. Instead of merely using age and sex as control variables, we computed partial correlations additionally controlling for the respective brain region's volume and NODDI parameters. We are confident that these measures are sufficient to avoid spurious findings merely driven by the effects of multicollinearity.

4. The kinetikor system didn't end up being used for anything but the earliest scans in the HCP study for technical reasons, so I would delete that sentence from the methods.

Response: We would like to thank the reviewer for pointing that out to us. The respective sentence was deleted from the methods section.

5. The number of diffusion directions should be stated for the HCP study.

Response: The number of diffusion directions is now included in the methods section.

6. It is not necessary to convert surface label files to volume files to compute overlap, however if the same surface is used for surface to volume mapping, it will not matter.

Response: The Human Connectome Project's multi-modal parcellation scheme and the Brodmann parcellation provided by the Van Essen Lab were both converted to volume files using FreeSurfer's fsaverage surface. Thus, as mentioned by the reviewer, this approach should not cause any problems.

7. Ideally the HCP parcellation's color scheme would be preserved, rather than using the random scheme currently used in Figure 1.

Response: We adapted the color scheme as requested.

Reviewer 4

The authors have adequately dealt with my initial critiques. The addition of a replication data set provides an additional check on the previously reported results, which improves the quality of the manuscript.

Response: We would like to thank the reviewer for acknowledging our efforts to further increase the quality of this manuscript. We agree that including data from the Human Connectome Project in order to validate our own results was a major step in the right direction. The findings obtained from both data sets are in substantial agreement with each other, even after rerunning our NODDI analyses with a different parameter of intrinsic diffusivity. Thus, we are convinced that our work features solid empirical evidence.

REVIEWERS' COMMENTS:

Reviewer #2 (Remarks to the Author):

The authors have agreed to incorporate my additional suggestions. One additional comment I have concerns the response to Reviewer 3 Query 2. It can be very valuable to include the original analysis as supplementary material. As the authors noted, the results are robust to the change made to the modelling. This in itself is important to be reported as part of this article.

Reviewer #3 (Remarks to the Author):

The authors have adequately addressed my concerns.

Response to Reviewers:
**Diffusion markers of dendritic density and arborization in gray matter
predict differences in intelligence**

Erhan Genç, Christoph Fraenz, Caroline Schlüter, Patrick Friedrich, Rüdiger Hossiep, Manuel
C. Voelke, Josef M. Ling, Onur Güntürkün and Rex E. Jung

Response to Reviewers

Reviewer 2

The authors have agreed to incorporate my additional suggestions. One additional comment I have concerns the response to Reviewer 3 Query 2. It can be very valuable to include the original analysis as supplementary material. As the authors noted, the results are robust to the change made to the modelling. This in itself is important to be reported as part of this article.

Response: We would like to thank the reviewer for this suggestion. However, we believe that adding our original analysis and all of its results as supplemental material would decrease the readability of the manuscript and divert the reader's attention away from our main message, i.e. the inverse relationship between intelligence and neurite microstructure in the cortex. Thus, we decided to only include the analysis preferred by Reviewer 3. Since reviewer comments to the authors and author rebuttal letters will be made available online as a supplementary peer review file, our use of different modelling approaches and the robust nature of our results will still be documented.

Reviewer 3

The authors have adequately addressed my concerns.

Response: We are very pleased to read that our last round of revisions was able to address all major concerns raised by the reviewer.